# Foundations of Symbolic Languages for Model Interpretability

**Marcelo Arenas**[1,4]**, Daniel Baez**[3]**, Pablo Barceló**[2,4]**, Jorge Pérez**[3,4]**, Bernardo Subercaseaux**[4,5]

[1] Department of Computer Science, PUC-Chile
[2] Institute for Mathematical and Computational Engineering, PUC-Chile
[3] Department of Computer Science, Universidad de Chile
[4] Millennium Institute for Foundational Research on Data, Chile
[5] Carnegie Mellon University, USA
[marenas, pbarcelo]@ing.puc.cl, jperez@dcc.uchile.cl, bsuberca@andrew.cmu.edu

## Abstract

Several queries and scores have been proposed to explain individual predictions made by ML models. Examples include queries based on "anchors", which are parts of an instance that are sufficient to justify its classification, and "feature-perturbation" scores such as SHAP. Given the need for flexible, reliable, and easy-to-apply interpretability methods for ML models, we foresee the need for developing declarative languages to naturally specify different explainability queries. We do this in a principled way by rooting such a language in a logic called FOIL, that allows for expressing many simple but important explainability queries, and might serve as a core for more expressive interpretability languages. We study the computational complexity of FOIL queries over classes of ML models often deemed to be easily interpretable: decision trees and more general decision diagrams. Since the number of possible inputs for an ML model is exponential in its dimension, tractability of the FOIL evaluation problem is delicate, but can be achieved by either restricting the structure of the models, or the fragment of FOIL being evaluated. We also present a prototype implementation of FOIL wrapped in a high-level declarative language, and perform experiments showing that such a language can be used in practice.

## 1 Introduction

**Context.** The degree of *interpretability* of a machine learning (ML) model seems to be intimately related with the ability to "answer questions" about it. Those questions can either be global (behavior of the model as a whole) or local (behavior regarding certain instances/features). Concrete examples of such questions can be found in the recent literature, including, e.g., queries based on "anchors", which are parts of an instance that are sufficient to justify its classification [4, 13, 16, 32], and numerical scores that measure the impact of the different features of an instance on its result [24, 31, 36].

It is by now clear that ML interpretability admits no silver-bullet [18], and that in many cases a combination of different queries may be the most effective way to understand a model's behavior. Also, model interpretability takes different flavors depending on the application domain one deals with. This naturally brings to the picture the need for general-purpose specification languages that can provide flexibility and expressiveness to practitioners specifying interpretability queries. An even more advanced requirement for these languages is to be relatively easy to use in practice. This tackles the growing need for bringing interpretability methods closer to users with different levels of expertise.

35th Conference on Neural Information Processing Systems (NeurIPS 2021).

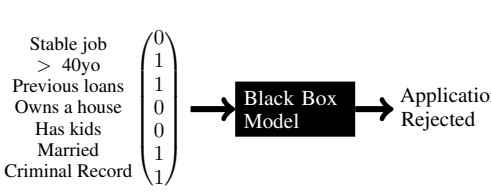



Stable job
> 40yo
Previous loans
Owns a house
Has kids
Married
Criminal Record

$\begin{pmatrix}0\\1\\1\\0\\0\\1\\1\end{pmatrix}$ → Black Box Model → Application Rejected

(a) Diagram of a particular loan decision.



```
> load("mlp.np") as MyModel;
> show features;
(stableJob, >40yo, prevLoan, ownsHouse,
hasKids, isMarried, crimRecord): Boolean
> show classes;
Rejected (0), Accepted (1)

> exists person,
    person.isMarried
    and not person.hasKids
    and MyModel(person) = Accepted;
YES
```

(b) Example of a possible concrete syntax for a language tailored for interpretability queries.

Figure 1: Example of a bank that uses a model to decide whether to accept loan applications considering binary features like "does the requester have a stable job" and "are they older than 40"?

One way in which these requirements can be approached in a principled way is by developing a *declarative* interpretability language, i.e., one in which users directly express the queries they want to apply in the interpretability process (and not how these queries will be evaluated). This is of course reminiscent of the path many other areas in computer science have followed, in particular by using languages rooted in formal logic; so has been the case, e.g., in data management [1], knowledge representation [3], and model checking [15]. One of the advantages of this approach is that logics have a well-defined syntax and clear semantics. On the one hand, this ensures that the obtained explanations are provably sound and faithful to the model, which avoids a significant drawback of several techniques for explaining models in which the explanations can be inaccurate, or require themselves to be further explained [33]. On the other hand, a logical root facilitates the theoretical study of the computational cost of evaluation and optimization for the queries in the language.

**Our proposal.** Our first contribution is the proposal of a logical language, called FOIL, in which many simple yet relevant interpretability queries can be expressed. We believe that FOIL can further serve as a basis over which more expressive interpretability languages can be built, and we propose concrete directions of research towards its expansion. In a nutshell, given a decision model $\mathcal{M}$ that performs classification over instances $\mathbf{e}$ of dimension $n$, FOIL can express properties over the set of all *partial* and *full* instances of dimension $n$. A partial instance $\mathbf{e}$ is a vector of dimension $n$ in which some features are undefined. Such undefined features take a distinguished value $\bot$. An instance is full if none of its features is undefined. The logic FOIL is simply first-order logic with access to two predicates on the set of all instances (partial or full) of dimension $n$: A unary predicate $\text{Pos}(\mathbf{e})$, stating that $\mathbf{e}$ is a full instance that $\mathcal{M}$ classifies as positive, and a binary predicate $\mathbf{e} \subseteq \mathbf{e}'$, stating that instance $\mathbf{e}'$ potentially fills some of the undefined features of instance $\mathbf{e}$; e.g., $(1, 0, \bot) \subseteq (1, 0, 1)$, but $(1, 0, \bot) \not\subseteq (1, 1, 1)$.

As an overview of our proposal, consider the case of a bank using a binary model to judge applications for loans. Figure 1a illustrates the problem with concrete features, and Figure 1b presents an example of a concrete interactive syntax. In Figure 1b, after loading and exploring the model, the interaction asks whether the model could give a loan to a person who is married and does not have kids. Assuming that the "Accepted" class is the positive one, this interaction can easily be formalized in FOIL by means of the query $\exists x \big( \text{Pos}(x) \wedge (\bot, \bot, \bot, \bot, 0, 1, \bot) \subseteq x \big)$.

**Theoretical contributions.** The *evaluation problem* for a fixed FOIL query $\varphi$ is as follows. Given a decision model $\mathcal{M}$, is it true that $\varphi$ is satisfied under the interpretation of predicates $\subseteq$ and $\text{Pos}$ defined above? An important caveat about this problem is that, in order to evaluate $\varphi$, we need to potentially look for an exponential number of instances, even if the features are Boolean, thus rendering the complexity of the problem infeasible in some cases. Think, for instance, of the query $\exists x \, \text{Pos}(x)$, which asks if $\mathcal{M}$ has at least one positive instance. Then this query is intractable for every class of models for which this problem is intractable; e.g., for the class of propositional formulas in CNF (notice that this is nothing but the *satisfiability problem* for the class at hand).

The main theoretical contribution of our paper is an in-depth study of the computational cost of FOIL on two classes of Boolean models that are often deemed to be "easy to interpret": decision trees and ordered binary decision diagrams (OBDDs) [10, 14, 19, 28, 33]. An immediate advantage of

these models over, say, CNF formulas, is that the satisfiability problem for them can be solved in polynomial time; i.e., the problem of evaluating the query $\exists x\text{Pos}(x)$ is tractable. Our study aims to (a) "measure" the degree of interpretability of said models in terms of the formal yardstick defined by the language FOIL; and (b) shed light on when and how some simple interpretability queries can be evaluated efficiently on these decision models.

We start by showing that, in spite of the aforementioned claims on the good level of interpretability for the models considered, there is a simple query in FOIL that is intractable over them. In fact, such an intractable query has a natural "interpretability" flavor, and thus we believe this proof to be of independent interest.

However, these intractability results should not immediately rule out the use of FOIL in practice. In fact, it is well known that a logic can be intractable in general, but become tractable in practically relevant cases. Such cases can be obtained by either restricting the syntactic fragment of the logic considered, or the structure of the models in which the logic is evaluated. We obtain positive results in both directions for the models we mentioned above. We explain them next.

*Syntactic fragments.* We show that queries in ∃FOIL, the existential fragment of FOIL, admit tractable evaluation over the models we study. However, this language lacks expressive power for capturing some interpretability queries of practical interest. We then introduce ∃FOIL$^+$, an extension of ∃FOIL with a finite set of unary universal queries from FOIL that is enough for expressing some relevant interpretability queries. We provide a characterization theorem for the tractability of ∃FOIL$^+$ over any class of Boolean decision models that reduces the tractability of this fragment to the tractability of two fixed and specific FOIL queries. Then we prove that said queries are tractable over perceptrons, which implies the tractability of ∃FOIL$^+$ for this model. Unfortunately, the evaluation of said queries is NP-hard for decision trees and OBDDs. Both the proof of tractability for perceptrons and intractability for decision trees and OBDDs are relatively simple, thus showing that the characterization theorem provides a useful technique for understanding which models are tractable for the evaluation of ∃FOIL$^+$.

*Structural restrictions.* We restrict the models allowed in order to obtain tractability of evaluation for *arbitrary* FOIL queries. In particular, we show that evaluation of $\varphi$, for $\varphi$ a fixed FOIL query, can be solved in polynomial time over the class of OBDDs as long as they are *complete*, i.e., any path from the root to a leaf of the OBDD tests every feature from the input, and have bounded *width*, i.e., there is a constant bound on the number of nodes of the OBDD in which a feature can appear.

**Practical implementation.** We designed FOIL with a minimal set of logical constructs and tailored for models with binary input features. These decisions are reasonable for a detailed theoretical analysis but may hamper FOIL usage in more general scenarios, in particular when models have (many) categorical or numerical input features, and queries are manually written by non-expert users. To tackle this we introduce a more user-friendly language with a high-level syntax (*à la* SQL in the spirit of the query in Figure 1b) that can be compiled into FOIL queries. Moreover, we present a prototype implementation that can be used to query decision trees trained in standard ML libraries by binarizing them into models (a subclass of binary decision diagrams) over which FOIL queries can be efficiently evaluated. We also test the performance of our implementation over synthetic and real data giving evidence of the usability of FOIL as a base for practical interpretabilty languages.

## 2   A Logic for Interpretability Queries

**Background.** An *instance* of dimension $n$, with $n \geq 1$, is a tuple $\mathbf{e} \in \{0,1\}^n$. We use notation $\mathbf{e}[i]$ to refer to the $i$-th component of this tuple, or equivalently, its $i$-th feature. Moreover, we consider an abstract notion of a model of dimension $n$, and we define it as a Boolean function $\mathcal{M} : \{0,1\}^n \to \{0,1\}$. That is, $\mathcal{M}$ assigns a Boolean value to each instance of dimension $n$, so that we focus on binary classifiers with Boolean input features. Restricting inputs and outputs to be Boolean makes our setting cleaner while still covering several relevant practical scenarios. We use notation $\dim(\mathcal{M})$ for the dimension of a model $\mathcal{M}$.

A *partial instance* of dimension $n$ is a tuple $\mathbf{e} \in \{0,1,\bot\}^n$. Intuitively, if $\mathbf{e}[i] = \bot$, then the value of the $i$-th feature is undefined. Notice that an instance is a particular case of a partial instance where all features are assigned value either $0$ or $1$. Given two partial instances $\mathbf{e}_1, \mathbf{e}_2$ of dimension $n$, we say that $\mathbf{e}_1$ is *subsumed* by $\mathbf{e}_2$ if for every $i \in \{1,\dots,n\}$ such that $\mathbf{e}_1[i] \neq \bot$, it holds that $\mathbf{e}_1[i] = \mathbf{e}_2[i]$.

That is, $\mathbf{e}_1$ is subsumed by $\mathbf{e}_2$ if it is possible to obtain $\mathbf{e}_2$ from $\mathbf{e}_1$ by replacing some unknown values. Notice that a partial instance $\mathbf{e}$ can be thought of as a compact representation of the set of instances $\mathbf{e}'$ such that $\mathbf{e}$ is subsumed by $\mathbf{e}'$, where such instances $\mathbf{e}'$ are called the *completions* of $\mathbf{e}$.

**Models.** A *binary decision diagram* (BDD [38]) over instances of dimension $n$ is a rooted directed acyclic graph $\mathcal{M}$ with labels on edges and nodes such that: (i) each leaf is labeled with **true** or **false**; (ii) each internal node (a node that is not a leaf) is labeled with a feature $i \in \{1, \ldots, n\}$; and (iii) each internal node has two outgoing edges, one labeled 1 and the another one labeled 0. Every instance $\mathbf{e} \in \{0, 1\}^n$ defines a unique path $\pi_{\mathbf{e}} = u_1 \cdots u_k$ from the root $u_1$ to a leaf $u_k$ of $\mathcal{M}$ such that: if the label of $u_i$ is $j \in \{1, \ldots, n\}$, where $i \in \{1, \ldots, k-1\}$, then the edge from $u_i$ to $u_{i+1}$ is labeled with $\mathbf{e}[j]$. Moreover, the instance $\mathbf{e}$ is positive, denoted by $\mathcal{M}(\mathbf{e}) = 1$, if the label of $u_k$ is **true**; otherwise the instance $\mathbf{e}$ is negative, which is denoted by $\mathcal{M}(\mathbf{e}) = 0$. A binary decision diagram $\mathcal{M}$ is *free* (FBDD) if for every path from the root to a leaf, no two nodes on that path have the same label. Besides, $\mathcal{M}$ is *ordered* (OBDD) if there exists a linear order $<$ on the set $\{1, \ldots, n\}$ of features such that, if a node $u$ appears before a node $v$ in some path in $\mathcal{M}$ from the root to a leaf, then $u$ is labeled with $i$ and $v$ is labeled with $j$ for features $i, j$ such that $i < j$. A *decision tree* is simply an FBDD whose underlying DAG is a tree. Finally, a perceptron $\mathcal{M}$ of dimension $n$ is a pair $(w, t)$ where $\boldsymbol{w} \in \mathbb{R}^n$ and $t \in \mathbb{R}$, and the classification of an instance $\mathbf{e} \in \{0, 1\}^n$ is defined as $\mathcal{M}(\mathbf{e}) = 1$ if and only if $\boldsymbol{w} \cdot \mathbf{e} \geq t$.

In this paper, we focus on the following classes of models: OBDD, the class of ordered BDDs, DTree, the class of decision trees, and Ptron, the class of perceptrons. None of these classes directly subsume the other: decision trees are not necessarily ordered, while the underlying DAG of an OBDD is not necessarily a tree. In fact, it is known that neither OBDDs can be compiled into polynomial-size decision trees nor decision trees into polynomial-size OBDDs [6, 17]. Perceptrons on the other hand can only model linear decision boundaries and thus are inherently less expressive than decision trees or OBDDs. It is also known that perceptrons cannot be compiled in polynomial time to decision trees or OBDDs unless $\mathrm{P} = \mathrm{NP}$ [4].

**The logic** FOIL. We consider first-order logic over a vocabulary consisting of a unary predicate Pos and a binary predicate $\subseteq$. This logic is called *first-order interpretability logic* (FOIL), and it is our reference language for defining conditions on models that we would like to reason about. In particular, predicate Pos is used to indicate the value of an instance in a model, while predicate $\subseteq$ is used to represent the subsumption relation among partial instances. In what follows, we show that many natural properties can be expressed in a simple way in FOIL, demonstrating the suitability of this language for the purpose of expressing explainability queries.

We assume familiarity with the syntax and semantics of first-order logic (see the appendix for a review of these concepts). In particular, given a vocabulary $\sigma$ consisting of relations $R_1, \ldots, R_\ell$, recall that a structure $\mathfrak{A}$ over $\sigma$ consists of a domain, where quantifiers are instantiated, and an interpretation for each relation $R_i$. Moreover, given a first-order formula $\varphi$ defined over the vocabulary $\sigma$, we write $\varphi(x_1, \ldots, x_k)$ to indicate that $\{x_1, \ldots, x_k\}$ is the set of free variables of $\varphi$. Finally, given a structure $\mathfrak{A}$ over the vocabulary $\sigma$ and elements $a_1, \ldots, a_k$ in the domain of $\mathfrak{A}$, we use $\mathfrak{A} \models \varphi(a_1, \ldots, a_k)$ to indicate that formula $\varphi$ is satisfied by $\mathfrak{A}$ when each variable $x_i$ is replaced by element $a_i$ ($1 \leq i \leq k$).

Our goal when introducing FOIL is to have a logic that allows to specify natural properties of models in a simple way. In this sense, we still need to define when a model $\mathcal{M}$ satisfies a formula in FOIL, as $\mathcal{M}$ is not a structure over the vocabulary $\{\text{Pos}, \subseteq\}$ (so we cannot directly use the notion of satisfaction of a formula by a structure). More precisely, assuming that $\dim(\mathcal{M}) = n$, the structure $\mathfrak{A}_{\mathcal{M}}$ associated to $\mathcal{M}$ is defined as follows. The domain of $\mathfrak{A}_{\mathcal{M}}$ is the set $\{0, 1, \bot\}^n$ of all partial instances of dimension $n$. An instance $\mathbf{e} \in \{0, 1\}^n$ is in the interpretation of predicate Pos in $\mathfrak{A}_{\mathcal{M}}$ if and only if $\mathcal{M}(\mathbf{e}) = 1$. Finally, a pair $(\mathbf{e}_1, \mathbf{e}_2)$ is in the interpretation of predicate $\subseteq$ in $\mathfrak{A}_{\mathcal{M}}$ if and only if $\mathbf{e}_1$ is subsumed by $\mathbf{e}_2$. Then, given a formula $\varphi(x_1, \ldots, x_k)$ in FOIL and partial instances $\mathbf{e}_1, \ldots, \mathbf{e}_k$ of dimension $n$, model $\mathcal{M}$ is said to *satisfy* $\varphi(\mathbf{e}_1, \ldots, \mathbf{e}_k)$, denoted by $\mathcal{M} \models \varphi(\mathbf{e}_1, \ldots, \mathbf{e}_k)$, if and only if $\mathfrak{A}_{\mathcal{M}} \models \varphi(\mathbf{e}_1, \ldots, \mathbf{e}_k)$.

**Evaluation problem.** FOIL is our main tool in trying to understand how interpretable is a class of models. In particular, the following is the main problem studied in this paper, given a class $\mathcal{C}$ of models and a formula $\varphi(x_1, \ldots, x_k)$ in FOIL.

| | |
|---|---|
| Problem: | EVAL($\varphi, \mathcal{C}$) |
| Input: | A model $\mathcal{M} \in \mathcal{C}$ of dimension $n$, and partial instances $\mathbf{e}_1, \dots, \mathbf{e}_k$ of dimension $n$ |
| Output: | YES, if $\mathcal{M} \models \varphi(\mathbf{e}_1, \dots, \mathbf{e}_k)$, and NO otherwise |

For example, assume that CNF, DNF are the classes of models given as propositional formulae in CNF and DNF, respectively. If $\varphi = \exists x\, \text{POS}(x)$, then EVAL($\varphi$, CNF) is NP-complete and EVAL($\varphi$, DNF) can be solved in polynomial time, as such problems correspond to the satisfiability problems for the propositional formulae in CNF and DNF, respectively.

Given a model $\mathcal{M}$, it is important to notice that the size of the structure $\mathfrak{A}_{\mathcal{M}}$ can be exponential in the size of $\mathcal{M}$. Hence, $\mathfrak{A}_{\mathcal{M}}$ is a theoretical construction needed to formally define the semantics of FOIL, but that should not be built when verifying in practice if a formula $\varphi$ is satisfied by $\mathcal{M}$. In fact, if we are aiming at finding tractable algorithms for FOIL-evaluation, then we need to design an algorithm that uses directly the encoding of $\mathcal{M}$ as a model (for example, as a binary decision tree) rather than as a logical structure. In other words, in order to evaluate a query $\varphi = \exists x\, \text{POS}(x)$ over a model $\mathcal{M}$ of dimension $n$, one could certainly iterate over all $2^n$ instances $\mathbf{e} \in \{0,1\}^n$ and evaluate $\mathcal{M}(\mathbf{e})$. This of course impractical for even small-dimensional data. Therefore, evaluating formulas without iterating over the entire space of (partial) instances is the main technical challenge behind the results presented in this paper.

## 3 Expressing Properties in the Logic

**Basic queries.** We provide some formulas in FOIL to gain more insight into this logic. Fix a model $\mathcal{M}$ of dimension $n$. We can ask whether $\mathcal{M}$ assigns value 1 to some instance by using FOIL-formula $\exists x\, \text{POS}(x)$. Similarly, formula $\exists y\, (\text{FULL}(y) \wedge \neg\text{POS}(y))$ can be used to check whether $\mathcal{M}$ assigns value 0 to some instance, where

$$\text{FULL}(x) \quad = \quad \forall y\, (x \subseteq y \to x = y) \tag{1}$$

is used to verify whether all values in $x$ are known (that is, $\mathcal{M} \models \text{FULL}(\mathbf{e})$ if and only if $\mathbf{e}$ is an instance). Notice that formula $\text{FULL}(y)$ has to be included in $\exists y\, (\text{FULL}(y) \wedge \neg\text{POS}(y))$ since $\mathcal{M} \models \neg\text{POS}(\mathbf{e})$ for each partial instance $\mathbf{e}$ with unknown values.

Given an instance $\mathbf{e}$ such that $\mathcal{M}(\mathbf{e}) = 1$, we can ask if the values assigned to the first two features are necessary to obtain a positive classification. Formally, define $\mathbf{e}_{\{1,2\}}$ as a partial instance such that $\mathbf{e}_{\{1,2\}}[1] = \mathbf{e}_{\{1,2\}}[2] = \bot$ and $\mathbf{e}_{\{1,2\}}[i] = \mathbf{e}[i]$ for every $i \in \{3, \dots, n\}$, and assume that

$$\varphi(x) \quad = \quad \forall y\, ((x \subseteq y \wedge \text{FULL}(y)) \to \text{POS}(y)).$$

If $\mathcal{M} \models \varphi(\mathbf{e}_{\{1,2\}})$, then the values assigned in $\mathbf{e}$ to the first two features are not necessary to obtain a positive classification. Notice that the use of unknown values in $\mathbf{e}_{\{1,2\}}$ is fundamental to reason about all possible assignments for the first two features, while keeping the remaining values of features unchanged. Besides, observe that a similar question can be expressed in FOIL for any set of features.

As before, we can ask if there is a completion of a partial instance $\mathbf{e}$ that is assigned value 1, by using FOIL-formula $\psi(x) = \exists y\, (x \subseteq y \wedge \text{FULL}(y) \wedge \text{POS}(y))$; that is, $\mathcal{M} \models \psi(\mathbf{e})$ if and only if there is an assignment for the unknown values of $\mathbf{e}$ that results in an instance classified positively.

**Minimal sufficient reasons.** Given an instance $\mathbf{e}$ and a partial instance $\mathbf{e}'$ that is subsumed by $\mathbf{e}$, consider the problem of verifying whether $\mathbf{e}'$ is a *sufficient reason* for $\mathbf{e}$ in the sense that every completion of $\mathbf{e}'$ is classified in the same way as $\mathbf{e}$ [4, 21, 34]. The following query expresses this:

$$\text{SR}(x, y) \quad = \quad \text{FULL}(x) \wedge y \subseteq x \wedge \forall z\, [(y \subseteq z \wedge \text{FULL}(z)) \to (\text{POS}(x) \leftrightarrow \text{POS}(z))], \tag{2}$$

given that $\mathcal{M} \models (\mathbf{e}, \mathbf{e}')$ if and only if $\mathbf{e}'$ is a sufficient reason for $\mathbf{e}$. Finally, it can also be expressed in FOIL the condition that $y$ is a *minimal* sufficient reason for $x$:

$$\text{MSR}(x, y) \quad = \quad \text{SR}(x, y) \wedge \forall z\, ((z \subseteq y \wedge \text{SR}(x, z)) \to z = y).$$

That is, $\mathcal{M} \models (\mathbf{e}, \mathbf{e}')$ if and only if $\mathbf{e}'$ is a sufficient reason for $\mathbf{e}$, and there is no partial instance $\mathbf{e}''$ such that $\mathbf{e}''$ is a sufficient reason for $\mathbf{e}$ and $\mathbf{e}''$ is properly subsumed by $\mathbf{e}'$. Minimal sufficient reasons have also been called PI-explanations or abductive explanations in the literature [20, 21, 26, 35].

**Bias detection queries.** Let us consider an elementary approach to fairness based on *protected* features, i.e., features from a set $P$ that should not be used for decision taking (e.g., gender, age,

marital status, etc). We use a formalization of this notion proposed in [16], while noting it does not capture many other forms of biases and unfairness [27], and is thus to be taken only as an example. Given a model $\mathcal{M}$ of dimension $n$, and a set of protected features $P \subseteq \{1, \ldots, n\}$, an instance $\mathbf{e}$ is said to be a *biased decision* of $\mathcal{M}$ if there exists an instance $\mathbf{e}'$ such that $\mathbf{e}$ and $\mathbf{e}'$ differ only on features from $P$ and $\mathcal{M}(\mathbf{e}) \neq \mathcal{M}(\mathbf{e}')$. A model $\mathcal{M}$ is *biased* if and only if there is an instance $\mathbf{e}$ that is a biased decision of $\mathcal{M}$. In what follows, we show how to encode queries relating to biased decisions in FOIL.

Let $S = \{1, \ldots, n\}$, and assume that $\mathbf{0}_S$ is an instance of dimension $n$ such that $\mathbf{0}_S[i] = 0$ for every $i \in S$, and $\mathbf{0}_S[j] = \bot$ for every $j \in \{1, \ldots, n\} \setminus S$. Moreover, define $\mathbf{1}_S$ in the same way but considering value 1 instead of 0, and define

$$\text{MATCH}(x, y, u, v) \quad = \quad \forall z \left[ (z \subseteq u \vee z \subseteq v) \rightarrow (z \subseteq x \leftrightarrow z \subseteq y) \right].$$

When this formula is evaluated replacing $u$ by $\mathbf{0}_S$ and $v$ by $\mathbf{1}_S$, it verifies whether $x$ and $y$ have the same value in each feature in $S$. More precisely, given a model $\mathcal{M}$ and instances $\mathbf{e}_1, \mathbf{e}_2$ of dimension $n$, we have that $\mathcal{M} \models \text{MATCH}(\mathbf{e}_1, \mathbf{e}_2, \mathbf{0}_S, \mathbf{1}_S)$ if and only if $\mathbf{e}_1[i] = \mathbf{e}_2[i]$ for every $i \in S$. Notice that the use of free variables $u$ and $v$ as parameters allows us to represent the matching of two instances in the set of features $S$, as, in fact, such matching is encoded by the formula $\text{MATCH}(x, y, \mathbf{0}_S, \mathbf{1}_S)$. The use of free variables as parameters is thus a useful feature of FOIL.

With the previous terminology, we can define a query

$$\begin{aligned} \text{BIASEDDECISION}(x, u, v) \quad = \quad \text{FULL}(x) \wedge \\ \exists y \left[ \text{FULL}(y) \wedge \text{MATCH}(x, y, u, v) \wedge (\text{POS}(x) \leftrightarrow \neg \text{POS}(y)) \right]. \end{aligned}$$

To understand the meaning of this formula, assume that $N = \{1, \ldots, n\} \setminus P$ is the set of non-protected features. When $\text{BIASEDDECISION}(x, u, v)$ is evaluated replacing $u$ by $\mathbf{0}_N$ and $v$ by $\mathbf{1}_N$, it verifies whether there exists an instance $y$ such that $x$ and $y$ have the same values in the non-protected features but opposite classification, so that $x$ is a biased decision. Hence, the formula

$$\text{BIASEDMODEL}(u, v) \quad = \quad \exists x \, \text{BIASEDDECISION}(x, u, v)$$

can be used to check whether a model $\mathcal{M}$ is biased with respect to the set $P$ of protected features, as $\mathcal{M}$ satisfies this property if and only if $\mathcal{M} \models \text{BIASEDMODEL}(\mathbf{0}_N, \mathbf{1}_N)$.

A query of the form $\exists x \left( \text{POS}(x) \wedge (\bot, \bot, \bot, \bot, 0, 1, \bot) \subseteq x \right)$ was included as an initial example in Section 1. According to the formal definition of FOIL, such a query corresponds to $\varphi(u) = \exists x \left( \text{POS}(x) \wedge u \subseteq x \right)$, and the desired answer is obtained when verifying whether $\varphi(\mathbf{e})$ is satisfied by a model, where $\mathbf{e}[1] = \mathbf{e}[2] = \mathbf{e}[3] = \mathbf{e}[4] = \bot$, $\mathbf{e}[5] = 0$, $\mathbf{e}[6] = 1$ and $\mathbf{e}[7] = \bot$. Again, notice that the use of free variables as parameters is an important feature of FOIL.

## 4 Limits to Efficient Evaluation

Several important interpretability tasks have been shown to be tractable for the decision models we study in the paper [4], which has justified the informal claim that they are "interpretable". But this does not mean that all interpretability tasks are in fact tractable for these models. We try to formalize this idea by studying the complexity of evaluation for queries in FOIL over them. We show next that the evaluation problem over the models studied in the paper can become intractable, even for some simple queries in the logic with a natural interpretability flavor. This intractability result is of importance, in our view, as it sheds light on the limits of efficiency for interpretability tasks over the models studied, and hence on the robustness of the folklore claims about them being "interpretable".

**Theorem 1.** *There exists a formula $\psi(x)$ in* FOIL *for which* $\text{EVAL}(\psi(x), \text{DTree})$ *and* $\text{EVAL}(\psi(x), \text{OBDD})$ *are* NP-*hard.*

This result tell us that there exists a concrete property expressible in FOIL that cannot be solved in polynomial time for decision trees and OBDDs (unless P = NP). In what follows, we describe this property, and how it is represented as a formula $\psi(x)$ in FOIL (the complete proof of Theorem 1 is provided in the appendix).

Assume that $x \subset y$ is the formula $x \subseteq y \wedge x \neq y$ that verifies whether $x$ is properly subsumed by $y$. We first define the following auxiliary predicates:

$$\begin{aligned} \text{ADJ}(x, y) \quad &= \quad x \subset y \wedge \neg \exists z \, (x \subset z \wedge z \subset y), \\ \text{DIFF}(x, y) \quad &= \quad \text{FULL}(x) \wedge \text{FULL}(y) \wedge x \neq y \wedge \exists z \, (\text{ADJ}(z, x) \wedge \text{ADJ}(z, y)). \end{aligned}$$

More precisely, $\textsc{Adj}(x, y)$ is used to check whether a partial instance $x$ is adjacent to a partial instance $y$, in the sense that $x$ is properly subsumed by $y$ and there is no partial instance $z$ such that $x$ is properly subsumed by $z$ and $z$ is properly subsumed by $y$. Moreover, $\textsc{Diff}(x, y)$ is used to verify whether two instances $x$ and $y$ differ exactly in the value of one feature. By using these predicates, we define the following notion of *stability* for an instance:

$$\textsc{Stable}(x) \quad = \quad \forall y \, [\textsc{Diff}(x, y) \to (\textsc{Pos}(x) \leftrightarrow \textsc{Pos}(y))].$$

That is, an instance $x$ is said to be stable if and only if any change in exactly one feature of $x$ leads to the same classification. Then the formula $\psi(x)$ in Theorem 1 is defined as follows:

$$\psi(x) \quad = \quad \exists y \, (x \subseteq y \wedge \textsc{Pos}(y) \wedge \textsc{Stable}(y)).$$

Hence, given a partial instance $x$, formula $\psi(x)$ is used to check if there is a completion of $x$ that is stable and positive. Theorem 1 states that checking this for decision trees and OBDDs is an intractable problem. Observe that the notion of stability used in $\psi(x)$ has a natural interpretability flavor: it identifies positive instances whose classification is not affected by the perturbation of a single feature. Note as well that the supposed interpretability of decision trees has already been questioned and nuanced in the literature [4, 23], to which this result contributes.

## 5 Tractable Restrictions

Theorem 1 tells us that evaluation of FOIL queries can be an intractable problem, but of course this does not completely rule out the applicability of the logic. In fact, as we show in this section one can obtain tractability by either restricting the analysis to a useful syntactic fragment of FOIL, or by considering a structural restriction on the class of models over which FOIL queries are evaluated.

### 5.1 A tractable fragment of FOIL

We present a fragment of FOIL that is simple enough to yield tractability, but which is at the same time expressive enough to encode natural interpretability problems. This is not a trivial challenge, though, as the proof of Theorem 1 shows intractability of queries in a syntactically simple fragment of FOIL (in fact, only two quantifier alternations suffice for the result to hold).

Our starting point in this search is $\exists\text{FOIL}$, which is the fragment of FOIL consisting of all formulas where no universal quantifier occurs and no existential quantifier appears under a negation (each such a formula can be rewritten into a formula of the form $\exists x_1 \cdots \exists x_k \, \alpha$, where $\alpha$ does not mention any quantifiers). Moreover, we consider the fragment $\forall\text{FOIL}$ of FOIL, which is defined in the same way as $\exists\text{FOIL}$ but exchanging the roles of universal and existential quantifiers. Then we show the following:

**Proposition 1.** *Let $\varphi$ be a query in $\exists\text{FOIL}$ or $\forall\text{FOIL}$. Then $\textsc{Eval}(\varphi, \textsf{DTree})$ and $\textsc{Eval}(\varphi, \textsf{OBDD})$ can be solved in polynomial time.*

However, the fragment $\exists\text{FOIL}$ has a limited expressive power since, for example, the predicate $\textsc{Full}(x)$ defined in (1) cannot be expressed in it (see Appendix B for a formal proof of this claim). To remedy this, we extend $\exists\text{FOIL}$ by including predicate $\textsc{Full}(x)$ and two other unary predicates that are common in interpretability queries. More precisely, let $\textsc{AllPos}(x)$ and $\textsc{AllNeg}(x)$ be unary predicates defined as follows:

$$\textsc{AllPos}(x) \quad = \quad \forall y \, \big( (x \subseteq y \wedge \textsc{Full}(y)) \to \textsc{Pos}(y) \big),$$
$$\textsc{AllNeg}(x) \quad = \quad \forall y \, \big( x \subseteq y \to \neg \textsc{Pos}(y) \big).$$

Then $\exists\text{FOIL}^+$ is defined as the fragment of FOIL consisting of all formulae where no universal quantifier occurs and no existential quantifier appears under a negation, and which are defined over the extended vocabulary $\{\textsc{Pos}, \subseteq, \textsc{Full}, \textsc{AllPos}, \textsc{AllNeg}\}$. In the same way, we define $\forall\text{FOIL}^+$ by exchanging the roles of universal and existential quantifiers. Notice that the formula defining the notion of sufficient reason in (2) is in $\forall\text{FOIL}^+$. Similarly, the notion of minimal sufficient reason introduced in Section 3 can be expressed in $\forall\text{FOIL}^+$:

$$\textsc{msr}(x, y) \ = \ \textsc{SR}(x, y) \wedge \forall u \, [(u \subseteq y \wedge u \neq y \wedge \textsc{Pos}(x)) \to \neg\textsc{AllPos}(u)] \wedge$$
$$\forall v \, [(v \subseteq y \wedge v \neq y \wedge \neg\textsc{Pos}(x)) \to \neg\textsc{AllNeg}(v)].$$

In what follows, we investigate the tractability of the fragments $\exists\text{FOIL}^+$ and $\forall\text{FOIL}^+$. In particular, in the case of $\exists\text{FOIL}^+$, we show that the tractability for a class of models $\mathcal{C}$ can be characterized in terms of the tractability in $\mathcal{C}$ of two specific queries in $\exists\text{FOIL}^+$:

$$\text{PARTIALALLPOS}(x, y, z) \;=\; \exists u\, [x \subseteq u \wedge \text{ALLPOS}(u) \wedge$$
$$\exists v\, (y \subseteq v \wedge u \subseteq v) \wedge \exists w\, (z \subseteq w \wedge u \subseteq w)],$$

and $\text{PARTIALALLNEG}(x, y, z)$ that is defined exactly as $\text{PARTIALALLPOS}(x, y, z)$ but replacing $\text{ALLPOS}(u)$ by $\text{ALLNEG}(u)$. More precisely, we have the following:

**Theorem 2.** *For every class $\mathcal{C}$ of models, the following conditions are equivalent: (a) $\text{EVAL}(\varphi, \mathcal{C})$ can be solved in polynomial time for each query $\varphi$ in $\exists\text{FOIL}^+$; (b) $\text{EVAL}(\text{PARTIALALLPOS}, \mathcal{C})$ and $\text{EVAL}(\text{PARTIALALLNEG}, \mathcal{C})$ can be solved in polynomial time.*

This theorem gives us a concrete way to study the tractability of $\exists\text{FOIL}^+$ over a class of models. Besides, as the negation of a query in $\forall\text{FOIL}^+$ is a query in $\exists\text{FOIL}^+$, Theorem 2 also provides us with a tool to study the tractability of $\forall\text{FOIL}^+$. In fact, it is possible to prove the following for the class Ptron of perceptrons.

**Proposition 2.** *The problems $\text{EVAL}(\text{PARTIALALLPOS}, \text{Ptron})$ and $\text{EVAL}(\text{PARTIALALLNEG}, \text{Ptron})$ can be solved in polynomial time.*

From this proposition and Theorem 2, it is possible to establish the following tractability results for $\exists\text{FOIL}^+$ and $\forall\text{FOIL}^+$.

**Corollary 1.** *Let $\varphi$ be a query in $\exists\text{FOIL}^+$ or $\forall\text{FOIL}^+$. Then $\text{EVAL}(\varphi, \text{Ptron})$ can be solved in polynomial time.*

In fact, a more general corollary holds: $\text{EVAL}(\varphi, \text{Ptron})$ is tractable as long as $\varphi$ is a *Boolean combination* of queries in $\exists\text{FOIL}^+$ (which covers the case of $\forall\text{FOIL}^+$). Unfortunately, these queries turn out to be intractable over decision trees and OBDDs.

**Proposition 3.** *Let $\mathcal{C}$ be OBDD or DTree. The problems $\text{EVAL}(\text{PARTIALALLPOS}, \mathcal{C})$ and $\text{EVAL}(\text{PARTIALALLNEG}, \mathcal{C})$ are NP-hard.*

## 5.2 A structural restriction ensuring tractability

We now look into the other direction suggested before, and identify a structural restriction on OBDDs that ensures tractability of evaluation for each query in FOIL. This restriction is based on the usual notion of *width* of an OBDD [5, 9]. An OBDD $\mathcal{M}$ over a set $\{1, \ldots, n\}$ of features is *complete* if each path from the root of $\mathcal{M}$ to one of its leaves includes every feature in $\{1, \ldots, n\}$. The *width* of $\mathcal{M}$, denoted by $\text{width}(\mathcal{M})$, is defined as the maximum value $n_i$ for $i \in \{1, \ldots, n\}$, where $n_i$ is the number of nodes of $\mathcal{M}$ labeled by feature $i$. Then, given $k \geq 1$, $k$-COBDD is defined as the class of complete OBDDs $\mathcal{M}$ such that $\text{width}(\mathcal{M}) \leq k$. By building on techniques from [9], we prove that:

**Theorem 3.** *Let $k \geq 1$ and query $\varphi$ in FOIL. Then $\text{EVAL}(\varphi, k\text{-COBDD})$ can be solved in polynomial time.*

# 6 Practical Implementation

The FOIL language has at least two downsides from a usability point of view. First, in FOIL every query is constructed using a minimal set of basic logical constructs. Moreover, the variables in queries are instantiated by feature vectors that may have hundreds of components. This implies that some simple queries may need fairly long and complicated FOIL expressions. Second, FOIL is designed to only work over models with binary input features. These downsides are a consequence of our design decisions that were reasonable for a detailed theoretical analysis but may hamper FOIL usage in more general scenarios, in particular when models have (many) categorical or numerical input features.

In this section, we describe a simple high-level syntax and implementation of a more user-friendly language (*à la* SQL) to query general decision trees, and we show how to compile it into FOIL queries to be evaluated over a suitable binarization of the queried model. As a whole, the pipeline requires several pieces that we explain in this section: (i) a working and efficient query-evaluation implementation of a fragment of FOIL over a suitable sub-class of Binary Decision Diagrams (BDDs),

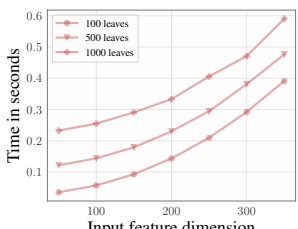
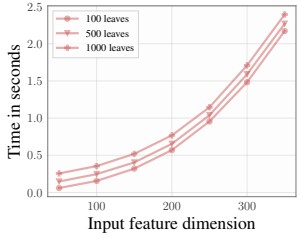

```
> exists student,
    student.age <= 18 and
    (student.internetAtHome or
     student.male) and
    goodGrades(student)
```

(a) Average time for 60 random FOIL queries over Decision Trees trained with random data.

(b) Maximum time for 60 random FOIL queries over Decision Trees trained with random data.

(c) Example of a query in our system executed over a model trained in the dataset in [29].

Figure 2: Execution time for FOIL queries and a high-level practical syntax.

(ii) a transformation from the high-level syntax to FOIL queries, and (iii) a transformation from a general decision tree to a BDD over which the FOIL query can be efficiently evaluated. We only present here the main ideas and intuitions of the implemented methods. A detailed exposition along with our implementation and a set of real examples can be found in the supplementary material.

## 6.1 Implementing and testing core FOIL

We implemented a version of the algorithm derived from Section 5.1 for evaluating existential and universal FOIL queries that is proven to work over a suitable sub-class of BDDs. The method receives a query as a plain text file and a BDD in JSON format. We tested the efficiency of our implementation varying three different parameters: the number of input features, the number of leaves of the decision tree, and the size of the input queries. We created a set of trees trained with random input data with input feature dimensions in the range $[10, 350]$, and of $100$, $500$ and $1000$ leaves (24 different decision trees). We note that the best performing decision trees over standard datasets [37] rarely contain more than 1000 total nodes [25], thus the trees that we tested can be considered of standard size. We created a set of random queries with 1 to 4 quantified variables, and a varying number of operators (60 different queries). We run every query 5 times over each tree, and averaged the execution time to obtain the running time of one case. From all our tests no case required more than 2.5 seconds for its complete evaluation with a total average execution time of $0.213$ seconds and standard deviation of $0.169$ in the whole dataset. Figure 2a shows the average time (average over different queries) for all settings. We observed that some queries where specially more time consuming than others. Figure 2b shows the maximum execution time over all queries for each setting. The most important factor when evaluating queries is the number of input features, which is consistent with a theoretical worst case analysis. All experiments where run on a personal computer with a 2.48GHz Intel N3060 processor and 2GB RAM. The exact details of the machine are presented in the supplementary material.

## 6.2 Interpretability symbolic queries in practice

**High-level features.** We designed and implemented a prototype system for user-friendly interpretability queries. Figure 2c shows a real example query that can be posed in our system for a model trained over the *Student Performance Data Set* [29]. Notice that our syntax allow named features, names for the target class (`goodGrades` in the example) and the comparison with numerical thresholds which goes beyond the FOIL formalization. Our current implementation allows for numerical and logical comparisons, as well as handy logical shortcuts such as `implies` and `iff`. Moreover we implemented a wrapper to directly import Decision Trees trained in the Scikit-learn [30] library.

**Binarizing models and queries.** One of the main issues when compiling these new queries into FOIL is how to binarize numerical features. Choi et al. [12] describe in extensive detail an approach to encode general decision trees into binary ones. The key observation is that one can separate numerical values into equivalence classes depending on the thresholds used by a decision tree. For example, assume a tree with an *age* feature that learns nodes with thresholds age $\leq 16$ and age $\leq 24$. It is clear that such a tree cannot distinguish an age $= 17$ from an age $= 19$. In general, every tree induces a finite number of equivalence classes for each numerical feature and one can take advantage of that to produce a binary version of the tree [12]. In our case, we also need to take the query into account. For instance, when evaluating a query with a condition `student.age <= 18`, ages 17 and

19 become distinguishable. Considering all these thresholds we have intervals $(-\infty, 16]$, $(16, 18]$, $(18, 24]$, $(24, \infty)$ and we can use four binary features to encode in which interval an age value lies. It is worth noting that this process creates extra artificial features, and thus, the decision tree that learned real thresholds needs to be binarized in the new feature space accordingly. One can show that a naive implementation would imply an exponential blow up in the size of the new tree. To avoid this our binarization process transforms the real-valued decision tree into a binary FBDD, over which we prove that our polynomial algorithms from Section 5.1 are still applicable.

**Performance tests.** We tested a set of 20 handcrafted queries over decision trees with up to 400 leaves trained for the Student Performance Data Set [29], which combines Boolean and numerical features. Our results show that natural queries can be evaluated over decision trees of standard size [25] in less than a second on a standard personal machine, thus validating the practical usability of our prototype.

## 7 Final Remarks and Future Work

In several aspects the logic FOIL is limited in expressive power for interpretability purposes. This was a design decision for this paper, in order to start with a "minimal" logic that would allow highlighting the benefits of having a declarative language for interpretability tasks, and at the same time allowing to carry out a clean theoretical analysis of its evaluation complexity. However, a genuinely practical declarative language should include other functionalities that allow more sophisticated queries to be expressed. As an example, consider the notion of SHAP-score [24] that has a predominant place in the literature on interpretability issues today. In a nutshell, for a decision model $\mathcal{M}$ with $\dim(\mathcal{M}) = n$ and instance $\mathbf{e} \in \{0, 1\}^n$, this score corresponds to a weighted sum of expressions of the form $\#\text{Pos}_S(\mathbf{e})$, for $S \subseteq \{1, \ldots, n\}$, where $\#\text{Pos}_S(\mathbf{e})$ is the number of instances $\mathbf{e}'$ for which $\mathcal{M}(\mathbf{e}') = 1$ and $\mathbf{e}'$ coincides with $\mathbf{e}$ over all features in $S$. Expressing this query, hence, requires extending FOIL with a recursive mechanism that permits to iterate over the subsets $S$ of $\{1, \ldots, n\}$, and a feature for counting the number of positive completions of a partial instance; e.g., in the form of a "numerical" query $\phi(x) := \#y.(x \subseteq y \wedge \text{Pos}(y))$. Logics of this kind abound in computer science logic (c.f., [2, 22]), and one could use all this knowledge in order to build a suitable extension of FOIL for dealing with this kind of interpretability tasks. One can also envision a language facilitating the comparison of different models by providing separate Pos predicates for each of them. Then, for example, one can ask whether two models are equivalent, or if they differ for a particular kind of instances. Such an extension can affect the complexity of evaluation in nontrivial ways.

Arguably, interpretability measures the degree in which *humans* can understand decisions made by *machines*. One of our main calls in this paper is to build more *symbolic* interpretability tools, and thus, make them closer to how humans reason about facts and situations. Having a symbolic high-level interpretability language to inspect ML models and their decisions is thus a natural and challenging way of pursuing this goal. We took a step further in this paper presenting theoretical and practical results, but several problems remain open. A particularly interesting one is whether a logical language can effectively interact with intrinsically non-symbolic models, and if so, what mechanisms could allow for practical tractability without sacrificing provable correctness.

## Acknowledgments and Disclosure of Funding

This work was partially funded by ANID - Millennium Science Initiative Program - Code ICN17_002. Arenas is funded by Fondecyt grant 1191337, while Barceló and Pérez are funded by Fondecyt grant 1200967.

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
