# Appendix

**Organization**   The supplementary material is organized as follows: Section A presents a brief review of the concepts concerning first-order logic that are used in our work. Section C presents a proof of Theorem 1, our negative result concerning decision trees and OBDDs, while Section D is devoted to our positive result. Section B proves that $\exists\text{FOIL}^+$ is strictly more expressive than $\exists\text{FOIL}$, justifying its independent study. Then, Section E is devoted to the tractability of $\exists\text{FOIL}$ and $\exists\text{FOIL}^+$; it includes proofs both for Theorem 2 and Proposition 2, which together imply the tractability of $\exists\text{FOIL}^+$ for perceptrons. Next, Section F presents a proof of Theorem 3, implying the full tractability of FOIL for a restricted class of OBDDs. Section G discusses details of the practical implementation, while Section H explains the the methodology of our experiments. Then, Section I discusses details of the high-level version we implemented, and also presents several examples of queries for the *Student Performance Data Set* which serve to show the usability of our implementation in practice. Finally Section J explains the binarization process for real-valued decision trees and high-level queries. A repository with code with our implementation for FOIL and the high-level syntax as well as examples and scripts to replicate our experiments can be found at

## A   Syntax and semantics of first-order logic

We review the definition of first-order logic (FO) over vocabularies consisting only of relations.

**Syntax of** FO.   A *vocabulary* $\sigma$ is a finite set $\{R_1, \ldots, R_m\}$, where each $R_i$ is a relation symbol with associated arity $n_i > 0$, for $i \in \{1, \ldots, m\}$. We assume the existence of a countably infinite set of variables $\{x, y, z, \ldots\}$, possibly with subscripts. The set of FO-*formulas over* $\sigma$ is inductively defined as follows.

1. If $x, y$ are variables, then $x = y$ is an FO-formula over $\sigma$.

2. If relation symbol $R \in \sigma$ has arity $n > 0$ and $x_1, \ldots, x_n$ are variables, then $R(x_1, \ldots, x_n)$ is an FO-formula over $\sigma$.

3. If $\varphi, \psi$ are FO-formulas over $\sigma$, then $(\neg\varphi)$, $(\varphi \vee \psi)$, and $(\varphi \wedge \psi)$ are FO-formulas over $\sigma$.

4. If $x$ is a variable and $\varphi$ is an FO-formula over $\sigma$, then $(\exists x\, \varphi)$ and $(\forall x\, \varphi)$ are FO-formulas over $\sigma$.

FO-formulas of type (1) and (2) are called *atomic*. A variable $x$ in FO-formula $\varphi$ appears *free*, if there is an occurrence of $x$ in $\varphi$ that is not in the scope of a quantifier $\exists x$ or $\forall x$. An FO-*sentence* is an FO-formula without free variables. We often write $\varphi(x_1, \ldots, x_k)$ to denote that $\{x_1, \ldots, x_k\}$ is the set of free variables of $\varphi$.

**Semantics of** FO.   FO-formulae over a vocabulary $\sigma$ are interpreted over $\sigma$-*structures*. Formally, a $\sigma$-structure is a tuple

$$\mathfrak{A} \;=\; \langle A,\, R_1^{\mathfrak{A}}, \cdots, R_m^{\mathfrak{A}}\rangle,$$

where $A$ is the *domain* of $\mathfrak{A}$, and for each relation symbol $R \in \sigma$ of arity $n$, we have that $R^{\mathfrak{A}}$ is an $n$-ary relation over $A$. We call $R_i^{\mathfrak{A}}$ the *interpretation* of $R_i$ in $\mathfrak{A}$.

Let $\varphi$ be an FO-formula over a vocabulary $\sigma$, and $\mathfrak{A}$ a $\sigma$-structure. Consider a mapping $\nu$ that associates an element in $A$ to each variable. We formally define the *satisfaction of* FO-*formula* $\varphi$ *over the pair* $(\mathfrak{A}, \nu)$, denoted by $(\mathfrak{A}, \nu) \models \varphi$, as follows.

1. If $\varphi$ is an atomic formula of the form $x = y$, then $(\mathfrak{A}, \nu) \models \varphi \Leftrightarrow \nu(t_1) = \nu(t_2)$.

2. If $\varphi$ is an atomic formula of the form $R(x_1, \ldots, x_n)$ for some $R \in \sigma$, then $(\mathfrak{A}, \nu) \models \varphi \Leftrightarrow (\nu(x_1), \ldots, \nu(x_n)) \in R^{\mathfrak{A}}$.

3. If $\varphi$ is of the form $(\neg\psi)$, then $(\mathfrak{A}, \nu) \models \varphi \Leftrightarrow (\mathfrak{A}, \nu) \not\models \psi$.

4. If $\varphi$ is of the form $(\psi \vee \psi')$, then $(\mathfrak{A}, \nu) \models \varphi$ iff $(\mathfrak{A}, \nu) \models \psi$ or $(\mathfrak{A}, \nu) \models \psi'$.

5. If $\varphi$ is of the form $(\psi \wedge \psi')$, then $(\mathfrak{A}, \nu) \models \varphi$ iff $(\mathfrak{A}, \nu) \models \psi$ and $(\mathfrak{A}, \nu) \models \psi'$.

6. If $\varphi$ is of the form $(\exists x\, \psi)$, then $(\mathfrak{A}, \nu) \models \varphi$ iff there exists $a \in A$ for which $(\mathfrak{A}, \nu[x/a]) \models \psi$. Here, $\nu[x/a]$ is a mapping that takes the same value as $\nu$ on every variable $y \neq x$, and takes value $a$ on $x$.

7. If $\varphi$ is of the form $(\forall x\, \psi)$, then $(\mathfrak{A}, \nu) \models \varphi$ iff for every $a \in A$ we have that $(\mathfrak{A}, \nu[x/a]) \models \psi$.

For an FO-formula $\varphi(x_1, \ldots, x_k)$ and assignment $\nu$ such that $\nu(x_i) = a_i$, for each $i \in \{1, \ldots, k\}$, we write $\mathfrak{A} \models \varphi(a_1, \ldots, a_k)$ to denote that $(\mathfrak{A}, \nu) \models \varphi$. If $\varphi$ is a sentence, we write simply $\mathfrak{A} \models \varphi$, as for any pair of mappings $\nu_1, \nu_2$ for the variables, it holds that $(\mathfrak{A}, \nu_1) \models \varphi$ iff $(\mathfrak{A}, \nu_2) \models \varphi$.

# B  Proof that the FULL predicate cannot be expressed in the existential fragment of FOIL

This proof requires some background in model theory. Namely, it uses the following ideas:

- Given a structure $\mathfrak{A}$ with domain $A$, a set $S \subseteq A$ induces a *sub-structure* $\mathfrak{A}'$ such that the domain of $\mathfrak{A}'$ is $A'$ and $R^{\mathfrak{A}'} = R^{\mathfrak{A}} \cap S^n$ for every relation $R \in \sigma$ of arity $n$.

- Let $\mathfrak{A}, \mathfrak{B}$ be two structures over a vocabulary $\sigma$ with domains $A$ and $B$, respectively. An *isomorphism* $f : A \to B$ between $\mathfrak{A}$ and $\mathfrak{B}$ satisfies the following property for every FO-formula $\varphi$ over $\sigma$, and every mapping $\nu$:

$$(\mathfrak{A}, \nu) \models \varphi \iff (\mathfrak{B}, f \circ \nu) \models \varphi$$

where $(f \circ \nu)$ is a mapping that associates $f(\nu(x))$ to each variable $x$.

- If $\varphi(x_1, \ldots, x_k)$ is an existential FO-formula over a vocabulary $\sigma$, $\mathfrak{A}$ is a $\sigma$-structure with domain $A$, $\mathfrak{A}'$ is an induced sub-structure of $\mathfrak{A}$ with domain $A'$, and $a_1, \ldots, a_k \in A'$:

$$\mathfrak{A}' \models \varphi(a_1, \ldots, a_k) \implies \mathfrak{A} \models \varphi(a_1, \ldots, a_k)$$

All these ideas are standard, and can be found for example in the reference book of Chang and Keisler [11].

We now proceed with the actual proof. For the sake of contradiction, assume that $\mathrm{FULL}(x)$ can be expressed in $\exists$FOIL. More precisely, assume that $\varphi(x)$ is a formula in $\exists$FOIL such that for every $n \geq 1$, every model $\mathcal{M}$ of dimension $n$, and every partial instance $\mathbf{e}$ of dimension $n$:

$$\mathfrak{A}_{\mathcal{M}} \models \varphi(\mathbf{e}) \quad \text{if and only if} \quad \mathbf{e} \text{ is an instance.} \tag{3}$$

Let $\mathcal{M}_1$ be a model of dimension 1 such that $\mathcal{M}_1(\mathbf{e}) = 0$ for every instance $\mathbf{e}$. Then we have that:

$$\mathfrak{A}_{\mathcal{M}_1} = \langle \{\bot, 0, 1\}, \mathrm{POS}^{\mathfrak{A}_{\mathcal{M}_1}}, \subseteq^{\mathfrak{A}_{\mathcal{M}_1}} \rangle,$$

where $\mathrm{POS}^{\mathfrak{A}_{\mathcal{M}_1}} = \emptyset$. Moreover, given condition 3, we also know that $\mathfrak{A}_{\mathcal{M}_1} \models \varphi((0))$. Let $\mathcal{M}_2$ be a model of dimension 2 such that $\mathcal{M}_2(\mathbf{e}) = 0$ for every instance $\mathbf{e}$. Then we have that:

$$\mathfrak{A}_{\mathcal{M}_2} = \langle \{\bot, 0, 1\}^2, \mathrm{POS}^{\mathfrak{A}_{\mathcal{M}_2}}, \subseteq^{\mathfrak{A}_{\mathcal{M}_2}} \rangle,$$

where $\mathrm{POS}^{\mathfrak{A}_{\mathcal{M}_2}} = \emptyset$. Moreover, let $\mathfrak{A}'$ be the sub-structure of $\mathfrak{A}_{\mathcal{M}_2}$ induced by the set of instances $\{(\bot, \bot), (0, \bot), (1, \bot)\}$. Then we have that function $f : \{(\bot), (0), (1)\} \to \{(\bot, \bot), (0, \bot), (1, \bot)\}$ defined as $f((x)) = (x, \bot)$ is an isomorphism from $\mathfrak{A}_{\mathcal{M}_1}$ to $\mathfrak{A}'$ such that $f((0)) = (0, \bot)$. Hence, given that $\varphi(x)$ is a formula in first-order logic and $\mathfrak{A}_{\mathcal{M}_1} \models \varphi((0))$, we conclude that $\mathfrak{A}' \models \varphi((0, \bot))$. Moreover, given that $\mathfrak{A}'$ is an induced sub-structure of $\mathfrak{A}_{\mathcal{M}_2}$ and $\varphi(x)$ is an existential formula in first-order logic, we have that $\mathfrak{A}_{\mathcal{M}_2} \models \varphi((0, \bot))$. Notice that this contradicts condition 3, as $(0, \bot)$ is not an instance.

# C  Proof of Theorem 1

Let us restate the theorem for the reader's convenience.

**Theorem 1.** *There exists a formula $\psi(x)$ in* FOIL *for which* EVAL($\psi(x)$, DTree) *and* EVAL($\psi(x)$, OBDD) *are* NP-*hard.*

*Proof.* We show that the problem is NP-hard by reducing from the satisfiability problem for propositional formulas in 3-CNF. We will in fact show that hardness holds already for the class DTree ∩ OBDD, which proves both cases a once. Let $\varphi = C_1 \wedge \cdots \wedge C_n$ be a propositional formula, where each $C_i$ is a disjunction of three literal and does not contain repeated or complementary literals. Moreover, assume that $\{x_1, \ldots, x_m\}$ is the set of variables occurring in $\varphi$, and the proof will use partial instances of dimension $n + m$. Notice that the last $m$ features of such a partial instance **e** naturally define a truth assignment for the propositional formula $\varphi$. More precisely, for every $i \in \{1, \ldots, n\}$, we use notation $\mathbf{e}(C_i) = 1$ to indicate that there is a disjunct $\ell$ of $C_i$ such that $\ell = x_j$ and $\mathbf{e}[n + j] = 1$, or $\ell = \neg x_j$ and $\mathbf{e}[n + j] = 0$, for some $j \in \{1, \ldots, m\}$. Furthermore, we say $\mathbf{e}(\varphi) = 1$ if $\mathbf{e}(C_i) = 1$ for every $i \in \{1, \ldots, n\}$.

We will build an *ordered decision tree* (thus belonging to DTree ∩ OBDD), over the natural ordering $1 < 2 < \cdots < n + m - 1 < n + m$. Let us denote this ordering with $\prec$ in order to avoid confusion. For each clause $C_i$ ($i \in \{1, \ldots, n\}$), let $\mathcal{T}_{C_i}$ be a decision tree of dimension $n + m$ (but that will only use features $n + 1, \ldots, n + m$) such that for every entity **e**: $\mathcal{T}_{C_i}(\mathbf{e}) = 1$ if and only $\mathbf{e}(C_i) = 1$. Moreover, we require each $\mathcal{T}_{C_i}$ to be ordered with respect to $\prec$, Notice that $\mathcal{T}_{C_i}$ can be constructed in constant time as it only needs to contain at most eight paths of depth 3. For example, assuming that $C = (x_1 \vee x_2 \vee x_3)$, a possible decision tree $\mathcal{T}_C$ is depicted in the following figure:

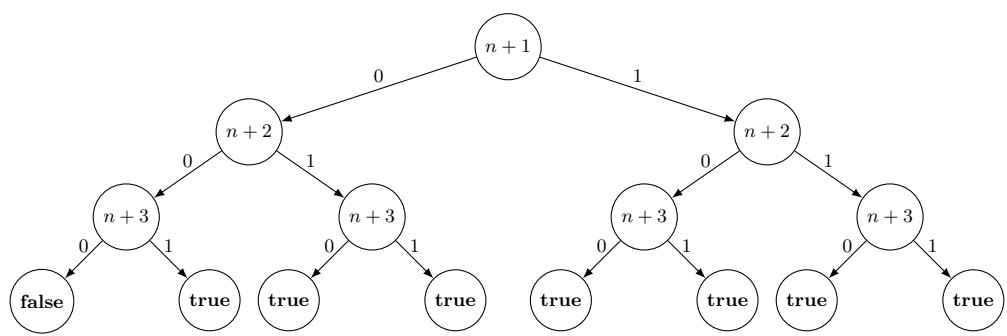

Moreover, define $\mathcal{T}_\varphi$ as the following decision tree, clearly ordered with respect to $\prec$:

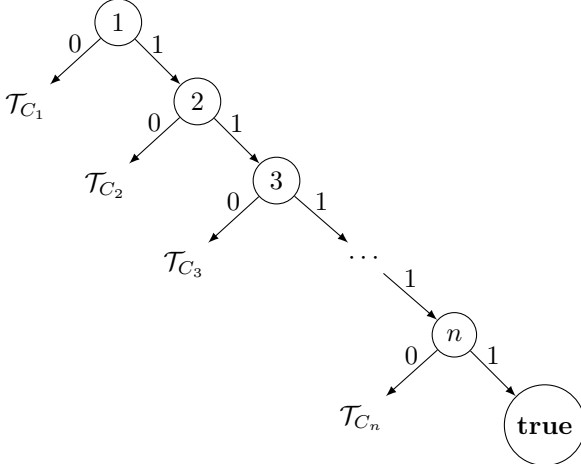

Finally, define **e** as a partial instance of dimension $n + m$ such that $\mathbf{e}[i] = 1$ for every $i \in \{1, \ldots, n\}$, and $\mathbf{e}[n+j] = \perp$ for every $j \in \{1, \ldots, m\}$, and define $\psi(x)$ as the following formula in FOIL (equiv-

alent to the formula presented in the body of the paper):

$$\psi(x) \;=\; \exists y \,(x \subseteq y \wedge \text{FULL}(y) \wedge \forall z \,((z \subseteq y \wedge \neg y \subseteq z) \;\rightarrow$$
$$\exists u \,(z \subseteq u \wedge \neg u \subseteq z \wedge \neg\text{FULL}(u)) \vee \forall v \,((z \subseteq v \wedge \neg v \subseteq z) \rightarrow \text{POS}(v)))). \quad (4)$$

Interestingly, $\psi(x)$ can be rewritten by using only two variables, which proves that an even more restricted fragment of FOIL is hard to evaluate. The following two-variable formula is equivalent to $\psi(x)$:

$$\exists y \,(x \subseteq y \wedge \text{FULL}(y) \wedge \forall x \,((x \subseteq y \wedge \neg y \subseteq x) \;\rightarrow$$
$$\exists y \,(x \subseteq y \wedge \neg y \subseteq x \wedge \neg\text{FULL}(y)) \vee \forall y \,((x \subseteq y \wedge \neg y \subseteq x) \rightarrow \text{POS}(y)))).$$

In what follows, we prove that $\varphi$ is satisfiable if and only if $\mathfrak{A}_{\mathcal{T}_\varphi} \models \psi(\mathbf{e})$, from which we conclude that the theorem holds.

($\Leftarrow$) Assume that $\mathfrak{A}_{\mathcal{T}_\varphi} \models \psi(\mathbf{e})$, and assume that $\mathbf{e}_1$ is a witness for the variable $y$, that is,

$$\mathfrak{A}_{\mathcal{T}_\varphi} \;\models\; \mathbf{e} \subseteq \mathbf{e}_1 \wedge \text{FULL}(\mathbf{e}_1) \wedge \forall z \,((z \subseteq \mathbf{e}_1 \wedge \neg \mathbf{e}_1 \subseteq z) \;\rightarrow$$
$$\exists u \,(z \subseteq u \wedge \neg u \subseteq z \wedge \neg\text{FULL}(u)) \vee \forall v \,((z \subseteq v \wedge \neg v \subseteq z) \rightarrow \text{POS}(v))).$$

In what follows, we show that $\mathbf{e}_1(\varphi) = 1$, from which we conclude that $\varphi$ is satisfiable. Fix an arbitrary $i \in \{1, \ldots, n\}$. Then, let $\mathbf{e}_2$ be a partial instance of dimension $n + m$ such that (i) $\mathbf{e}_2[i] = \bot$; (ii) $\mathbf{e}_2[j] = 1$ for each $j \in \{1, \ldots, n\}$ with $j \neq i$, and (iii) $\mathbf{e}_2[n + k] = \mathbf{e}_1[n + k]$ for each $k \in \{1, \ldots, m\}$. Then given that $(\mathbf{e}_2 \subseteq \mathbf{e}_1 \wedge \neg \mathbf{e}_1 \subseteq \mathbf{e}_2)$, we have that:

$$\mathfrak{A}_{\mathcal{T}_\varphi} \;\models\; \exists u \,(\mathbf{e}_2 \subseteq u \wedge \neg u \subseteq \mathbf{e}_2 \wedge \neg\text{FULL}(u)) \vee$$
$$\forall v \,((\mathbf{e}_2 \subseteq v \wedge \neg v \subseteq \mathbf{e}_2) \rightarrow \text{POS}(v)).$$

Therefore, given that $\mathbf{e}_2$ assigns value $\bot$ to exactly one feature, we conclude that:

$$\mathfrak{A}_{\mathcal{T}_\varphi} \;\models\; \forall v \,((\mathbf{e}_2 \subseteq v \wedge \neg v \subseteq \mathbf{e}_2) \rightarrow \text{POS}(v)). \quad (5)$$

Define $\mathbf{e}_3$ as an instance of dimension $n + m$ such that $\mathbf{e}_3[i] = 0$, $\mathbf{e}_3[j] = 1$ for each $j \in \{1, \ldots, n\}$ with $j \neq i$, and $\mathbf{e}_3[n + k] = \mathbf{e}_2[n + k]$ for each $k \in \{1, \ldots, m\}$. Then, by considering that $(\mathbf{e}_2 \subseteq \mathbf{e}_3 \wedge \neg \mathbf{e}_3 \subseteq \mathbf{e}_2)$ holds, we conclude from 5 that $\text{POS}(\mathbf{e}_3)$ holds. Therefore, given that $\mathbf{e}_3[i] = 0$ and $\mathbf{e}_3[j] = 1$ for each $j \in \{1, \ldots, n\}$ with $j < i$, we have that $\mathcal{T}_{C_i}(\mathbf{e}_3) = 1$, from which we deduce that $\mathcal{T}_{C_i}(\mathbf{e}_1) = 1$, since $\mathbf{e}_1[n + j] = \mathbf{e}_2[n + j] = \mathbf{e}_3[n + j]$ for every $j \in \{1, \ldots, m\}$. As $i$ is an arbitrary element in the set $\{1, \ldots, n\}$, we conclude that $\mathbf{e}_1(C_i) = 1$ for every $i \in \{1, \ldots, n\}$ and, thus, $\mathbf{e}_1(\varphi) = 1$, which was to be shown.

($\Rightarrow$) Assume that $\varphi$ is satisfiable, and let $\sigma$ be a truth assignment such that $\sigma(\varphi) = 1$. Moreover, define an instance $\mathbf{e}_1$ of dimension $n + m$ such that $\mathbf{e}_1[i] = 1$ for each $i \in \{1, \ldots, n\}$ and $\mathbf{e}_1[n + j] = \sigma(x_j)$ for each $j \in \{1, \ldots, m\}$. Then we have that $\mathbf{e}_1(\varphi) = 1$, $\mathbf{e} \subseteq \mathbf{e}_1$ and $\text{FULL}(\mathbf{e}_1)$ hold. Next we show that:

$$\mathfrak{A}_{\mathcal{T}_\varphi} \;\models\; \forall z \,((z \subseteq \mathbf{e}_1 \wedge \neg \mathbf{e}_1 \subseteq z) \;\rightarrow$$
$$\exists u \,(z \subseteq u \wedge \neg u \subseteq z \wedge \neg\text{FULL}(u)) \vee \forall v \,((z \subseteq v \wedge \neg v \subseteq z) \rightarrow \text{POS}(v))),$$

from which we conclude that $\mathfrak{A}_{\mathcal{T}_\varphi} \models \psi(\mathbf{e})$. Let $\mathbf{e}_2$ be a partial instance of dimension $n + m$ such that $(\mathbf{e}_2 \subseteq \mathbf{e}_1 \wedge \neg \mathbf{e}_1 \subseteq \mathbf{e}_2)$ holds. We need to prove that:

$$\mathfrak{A}_{\mathcal{T}_\varphi} \;\models\; \exists u \,(\mathbf{e}_2 \subseteq u \wedge \neg u \subseteq \mathbf{e}_2 \wedge \neg\text{FULL}(u)) \vee$$
$$\forall v \,((\mathbf{e}_2 \subseteq v \wedge \neg v \subseteq \mathbf{e}_2) \rightarrow \text{POS}(v)).$$

Notice that $\mathbf{e}_2$ assigns value $\bot$ to at least one feature in $X$, since $(\mathbf{e}_2 \subseteq \mathbf{e}_1 \wedge \neg \mathbf{e}_1 \subseteq \mathbf{e}_2)$ holds. If $\mathbf{e}_2$ assigns value $\bot$ to at least two features, then clearly $\mathfrak{A}_{\mathcal{T}_\varphi} \models \exists u \,(\mathbf{e}_2 \subseteq u \wedge \neg u \subseteq \mathbf{e}_2 \wedge \neg\text{FULL}(u))$. Hence, assume that $\mathbf{e}_2$ assigns value $\bot$ to exactly one feature, and consider the following cases.

- If $\mathbf{e}_2[n+j] = \bot$ for some $j \in \{1, \ldots, m\}$. Then for every partial instance $\mathbf{e}_3$ of dimension $n+m$ such that $(\mathbf{e}_2 \subseteq \mathbf{e}_3 \wedge \neg\mathbf{e}_3 \subseteq \mathbf{e}_2)$ holds, we have that $\mathbf{e}_3[i] = 1$ for every $i \in \{1, \ldots, n\}$. Therefore, from the definition of $\mathcal{T}_\varphi$, we conclude that $\text{POS}(\mathbf{e}_3)$ holds. Thus, we have that $\mathfrak{A}_{\mathcal{T}_\varphi} \models \forall v ((\mathbf{e}_2 \subseteq v \wedge \neg v \subseteq \mathbf{e}_2) \to \text{POS}(v))$.

- If $\mathbf{e}_2[i] = \bot$ for some $i \in \{1, \ldots, n\}$. Then assume that $\mathbf{e}_3$ is a partial instance of dimension $n+m$ such that $(\mathbf{e}_2 \subseteq \mathbf{e}_3 \wedge \neg\mathbf{e}_3 \subseteq \mathbf{e}_2)$ holds. If $\mathbf{e}_3[i] = 1$, then we have that $\mathbf{e}_3[j] = 1$ for every $j \in \{1, \ldots, n\}$, and we conclude by definition of $\mathcal{T}_\varphi$ that $\text{POS}(\mathbf{e}_3)$ holds. If $\mathbf{e}_3[i] = 0$, then we conclude that $\mathcal{T}_\varphi(\mathbf{e}_3) = \mathcal{T}_{C_i}(\mathbf{e}_3)$, since $\mathbf{e}_3[j] = 1$ for every $j \in \{1, \ldots, n\}$ such that $j < i$. Given that $\mathbf{e}_1$, $\mathbf{e}_2$ and $\mathbf{e}_3$ only differ in the value of $f_i$, we have that $\mathbf{e}_3[n+k] = \mathbf{e}_2[n+k] = \mathbf{e}_1[n+k]$ for every $k \in \{1, \ldots, m\}$, so that $\mathcal{T}_{C_i}(\mathbf{e}_3) = \mathcal{T}_{C_i}(\mathbf{e}_1)$. But then given that $\mathbf{e}_1(\varphi) = 1$, we know that $\mathcal{T}_{C_i}(\mathbf{e}_1) = 1$, which implies that $\mathcal{T}_\varphi(\mathbf{e}_3) = \mathcal{T}_{C_i}(\mathbf{e}_3) = \mathcal{T}_{C_i}(\mathbf{e}_1) = 1$. We conclude again that $\text{POS}(\mathbf{e}_3)$ holds, from which we deduce that $\mathfrak{A}_{\mathcal{T}_\varphi} \models \forall v ((\mathbf{e}_2 \subseteq v \wedge \neg v \subseteq \mathbf{e}_2) \to \text{POS}(v))$.

This concludes the proof of the theorem.

$\square$

## D  Proof of Proposition 1

Let us restate and prove the corresponding proposition.

**Proposition 1.** *Let $\varphi$ be a query in $\exists$FOIL or $\forall$FOIL. Then $\text{EVAL}(\varphi, \text{DTree})$ and $\text{EVAL}(\varphi, \text{OBDD})$ can be solved in polynomial time.*

*Proof.* We will prove this for the more general class of FBDD, that contains both DTree and OBDD. Assume that the input formula is of the form $\varphi = \exists x_1, \cdots, \exists x_k \psi(x_1, \ldots, x_k)$, with $\psi$ quantifier-free, and let $\mathcal{M}$ be the input model with $\dim(\mathcal{M}) = n$. Our algorithm will try to construct a valuation $\mathbf{e}_1, \ldots, \mathbf{e}_k$ of the variables of $\varphi$ such that $\mathcal{M} \models \psi(\mathbf{e}_1, \ldots, \mathbf{e}_k)$, and if this fails, it will be certain that no satisfying valuation exists.

We assume as well the input formula has constants but no free-variables as if the input instance has free variable we can simply replace them by the partial instances $\mathbf{e}_i$ supplied in the input. Let $V = \{\boldsymbol{x}_1, \ldots, \boldsymbol{x}_k\}$ be the variables mentioned in $\varphi$, of which there is only a constant number as $\varphi$ is fixed. Let $E = \{\mathbf{e}_1, \ldots, \mathbf{e}_k\}$ be their corresponding undetermined instances.

For each element in the domain of $\mathfrak{A}_{\mathcal{M}}$, that is, for each tuple in $\{0, 1, \bot\}^n$, we define its *type* as the set of unary predicates of $\exists$FOIL that it satisfies when interpreted over $\mathfrak{A}_{\mathcal{M}}$. In the case of $\exists$FOIL, this set corresponds either to $\{\text{POS}\}$ or to $\{\neg\text{POS}\}$, but we present the general strategy as it can be used for bigger fragments of FOIL, as shown later in the proof of Theorem 2.

Let $\mathcal{T}$ be the set of types which is of course a fixed set independent of $\mathcal{M}$. We will guess the type of each instance $\mathbf{e} \in E$. More formally, we can iterate over all type assignments $\tau : E \to \mathcal{T}$ as there is only fixed number of them. Similarly, we can define a *containment assignment* $\gamma$ as an assignment of all the ordered pairs $(\mathbf{e}_i, \mathbf{e}_j)$ to $\{0, 1\}$, with the meaning that $\mathbf{e}_i \subseteq \mathbf{e}_j$ iff $\gamma(\mathbf{e}_i, \mathbf{e}_j) = 1$. Such an assignment is said to be *possible* only if it holds the properties of a partial order. Given a possible containment assignment $\gamma$, we can interpret it as a pair of sets

$$P = \{(\mathbf{e}_i, \mathbf{e}_j) \mid \gamma(\mathbf{e}_i, \mathbf{e}_j) = 1\} \quad ; \quad N = \{(\mathbf{e}_i, \mathbf{e}_j) \mid \gamma(\mathbf{e}_i, \mathbf{e}_j) = 0\}$$

Note as well that there is a constant number of possibilities for the pair $P, N$. Because the formula $\varphi$ is existential, if there is an determinization of $E$ that models $\varphi$, then there is a pair $(\tau, \gamma = (P, N))$ where $\tau$ is a possible type assignment and $\gamma = (P, N)$ is a possible containment assignment, such that $E$ is *consistent* with both $\tau$ and $\gamma$. More precisely, $E$ is consistent with $\tau$ and $\gamma$ iff:

- For every $\mathbf{e} \in E$ and every unary predicate $\rho$,

$$\rho \in \tau(\mathbf{e}) \iff \mathbf{e} \in \rho^{\mathfrak{A}_{\mathcal{M}}}$$

- For every pair $\mathbf{e}_i, \mathbf{e}_j \in E$,

$$\gamma(\mathbf{e}_i, \mathbf{e}_j) = 1 \iff (\mathbf{e}_i, \mathbf{e}_j) \in \subseteq^{\mathfrak{A}_{\mathcal{M}}}$$

We can afford to iterate over the constantly many pairs $(\tau, \gamma)$, and for each pair $(\tau, \gamma)$ it is trivial to decide whether $\varphi$ gets satisfied under said assignments (simply by replacing every atomic term in $\varphi$ by the value assigned to it by $\tau$ or $\gamma$). Therefore, in order to prove the whole theorem, it is enough to design a polynomial time algorithm that decides whether there determinization of $E$ that is a consistent with a given pair $(\tau, \gamma)$. More precisely, proving the next claim will be enough to conclude our proof.

**Claim 1.** *Given a pair $(\tau, \gamma = (P, N))$, one can check in polynomial time whether there is a determinization of $E$ that is consistent with $(\tau, \gamma)$.*

*Proof of Claim 1.* First, as the desired determinization $E$ must be consistent with $N$, it must hold that for every fact $(\mathbf{e}_i, \mathbf{e}_j) \in N$, there is an index $1 \leq k \leq n$ such that $\mathbf{e}_i[k] \neq \bot$ and $\mathbf{e}_i[k] \neq \mathbf{e}_j[k]$. We can afford to guess, for each of the constantly many facts $(\mathbf{e}_i, \mathbf{e}_j) \in N$, an index $k$ and the values of $\mathbf{e}_i[k], \mathbf{e}_j[k]$, that certify the fact. After said guesses have been made, we can assume a set $F$ of guessed facts of the form $\mathbf{e}[k] = \alpha$, with $\alpha \in \{0, 1, \bot\}$. Then, for every fact in $F$ of the form $\mathbf{e}[k] = \beta$, with $\beta \in \{0, 1\}$, we include in $F$ all facts of the form $\mathbf{e}'[k] = \beta$ for every $\mathbf{e}'$ such that $(\mathbf{e}, \mathbf{e}') \in P$. Also, for every fact in $F$ of the form $\mathbf{e}[k] = \bot$, we include $\mathbf{e}'[k] = \bot$ for every $\mathbf{e}'$ such that $(\mathbf{e}', \mathbf{e}) \in P$. As any determinization of $E$ respecting $F$ will at least be consistent with $N$, it remains only to check whether there is an interpretation $E$ respecting to $F$ that is consistent with $\tau$ and $P$.

If $F$ fully determines some predicates that a certain instance $\mathbf{e} \in E$ must satisfy, for example because $F$ contains facts $\mathbf{e}[k] = \beta \in \{0, 1\}$ for all $1 \leq k \leq n$ and thus we know that $\mathbf{e}$ must be a full instance, we can check whether $\text{POS}(\mathbf{e})$ and if that holds reject immediately if $\text{POS}(\mathbf{e}) \notin \tau(\mathbf{e})$. Therefore, we can safely assume this is not the case, and that $\tau$ is not directly contradicted by $F$. We thus modify the undetermined instances $\mathbf{e}_1, \ldots, \mathbf{e}_k$ according to $F$. Let us now interpret $P$ as a directed acyclic graph $G$ obtained in the following way: (i) create a node for every instance $\mathbf{e} \in E$, (ii) create an edge $\mathbf{e} \to \mathbf{e}'$ iff $(\mathbf{e}, \mathbf{e}') \in P$, (iii) collapse strongly connected components to a single node. Note that, as strongly connected components before the last step correspond to instances that must be equal, we can think of them as a single instance, because forcefully $\tau$ must assign the same to each of them. We can now view our problem as that of determinizing every node in a DAG $G$, in such a way that the containment dictated by the graph is satisfied, and so is $\tau$.

If $G$ has multiple connected components (it will only have a constant number of them), it is easy to see that we can simply make the check for each of them separately, and return that the instance is positive if every connected component holds the check. This is because different connected components do not share instances $\mathbf{e}$, and thus a determinization of a connected component is always compatible with the determinization of another connected component. As a consequence, our problem is now even smaller; we need to show that it is possible to determine in polynomial time if the undetermined components in each node of a given connected DAG $G$ can be assigned values that are consistent with given assignments $\tau$ and $P$, assuming the guessed facts $F$.

We now show a direct simple algorithm for this problem:

1. Choose an arbitrary topological ordering $\phi$ of $G$.

2. Iterate over the nodes according to $\phi$, and for each node $\mathbf{e}$ do the next step.

3. If $\text{POS} \in \tau(\mathbf{e})$, go to step 4., otherwise go to 5.

4. We determinize $\mathbf{e}$ in an arbitrary way that is accepted by $\mathcal{M}$. This is easily done in polynomial time for FBDDs; it is enough to prune the edges of the FBDD that contradict a defined feature in $\mathbf{e}$, and then find any positive leaf of the resulting model. Take $\mathbf{e}$ to be the next node according to $\phi$ and go back to 3. If there is no next node, go to 6.

5. Assign every undetermined component of $\mathbf{e}$ to $\bot$, as that does not restrict any future choices while ensuring that $\text{POS} \notin \tau(\mathbf{e})$. Take $\mathbf{e}$ to be the next node according to $\phi$ and go back to 3. If there is no next node, go to 6.

6. Now that nodes have no undetermined components, check that every fact dictated by $\tau$ is true for the values that have been determined. If all the facts are correctly satisfied, return Yes, otherwise return No.

It is clear that, if the preceding algorithm returns Yes, then it is correct, as it has a concrete determinization consistent with $\tau$, and it must be consistent with $P$ as every undetermined component that is assigned $0$ or $1$ its propagated to the successors in the graph. It only remains to justify that it is correct when it returns No. Assume, looking for a contradiction, that the algorithm returns No but there actually exists a determinization $B$ of $E$ that is consistent with $\tau$ and $P$, assuming the guessed facts $F$. Let $A$ be the determinization that the algorithm tested in step 6, and let $i$ be the first node according to $\phi$, the choice of the algorithm in step 1, such that $A(\phi_i) \neq B(\phi_i)$. Such an index must exists because $A$ must differ from $B$. Among all determinizations that are consistent with $\tau$ and $P$, let $B'$ be the one that maximizes the index $i$ of its first difference with $A$. Then, let $\mathbf{e}$ be $i$-th node according to $\phi$, and thus the first node where $A$ and $B'$ differ. If $\text{POS} \in \tau(\mathbf{e})$, the algorithm determinized $\mathbf{e}$ in an arbitrary way that makes $\mathbf{e}$ a positive instance. But then, as $\mathbf{e}$ is positive (and therefore a full instance), it cannot have any successors in $G$, and thus if we let $B'' := B'$ except for $B''(\mathbf{e}) := A(\mathbf{e})$, then $B''$ must also be consistent with $\tau$, which contradicts the maximality of $i$. If $\text{POS} \notin \tau(\mathbf{e})$ we have two cases, either $B'(\mathbf{e})$ is a full instance or not. If it is, then again it has no successors in $G$, so it must be that the inconsistency is that the algorithm determinized $\mathbf{e}$ in a way that makes it a positive instance. This is clearly not possible, as the only step in the algorithm that introduces values different from $\bot$, and thus that makes feasible for $\mathbf{e}$ to be a positive instance, is step 4, which occurs exactly when $\text{POS} \in \tau(\mathbf{e})$. It remains to see the case where $B'(\mathbf{e})$ is not a full instance. Assume $j$ is the first component for which $A(\mathbf{e})[j] \neq B'(\mathbf{e})[j]$. If $A(\mathbf{e})[j] = \bot$, then note that every successor of $B'(\mathbf{e})$ is also a successor of $A(\mathbf{e})$ and thus if $B'(\mathbf{e})$ is consistent with $\tau$, then so is $A(\mathbf{e})$. This implies the inconsistency in $\tau$ must appear later in $\phi$, and thus we can again take $B''$ equal to $B'$ except for $B''(\mathbf{e}) := A(\mathbf{e})$ which will contradict the maximality of $i$. If $A(\mathbf{e})[j] \neq \bot$, then said value need to come from $F$, as the algorithm only introduces the value $\bot$ for instances where $\text{POS} \notin \tau(\mathbf{e})$, which means that $B'(\mathbf{e})[j] = A(\mathbf{e})[j]$, as $B'$ must also respect $F$, which contradicts the minimality of $j$.

$\square$

As the preceding claim has been proved, and there are constantly many pairs $(\tau, \gamma)$ to consider, there is a polynomial time algorithm for the whole problem. $\square$

# E   Proof of Theorem 2, Proposition 2 and Proposition 3

Before the proofs, let us gain a better understanding on the PARTIALALLPOS and PARTIALALLNEG formulas. Recall that

$$\text{PARTIALALLPOS}(x, y, z) \;=\; \exists u \,[x \subseteq u \wedge \text{ALLPOS}(u) \,\wedge$$
$$\exists v \,(y \subseteq v \wedge u \subseteq v) \wedge \exists w \,(z \subseteq w \wedge u \subseteq w)] \quad (6)$$

We will prove that this query captures an important computational problem. Let us introduce a fourth kind of value: $\Diamond$, so we now define *undetermined instances* as tuples in $\{0, 1, \bot, \Diamond\}^n$ for some $n \geq 1$. A component with value $\Diamond$ is said to be *undetermined*. Given an undetermined instance $\mathbf{e}$ of dimension $n$, we say that a partial instance $\mathbf{e}'$ of dimension $n$ is a *determinization* of $\mathbf{e}$ if for $\mathbf{e}$ matches $\mathbf{e}'$ in every component that is not undetermined. Note that $\mathbf{e}$ cannot have undetermined components as it is a partial instance (i.e., a tuple in $\{0, 1, \bot\}^n$).

Consider now the following computational problem:

| | |
|---|---|
| Problem: | DETERMINIZATIONALLPOS($\mathcal{C}$) |
| Input: | A model $\mathcal{M} \in \mathcal{C}$ of dimension $n$, and an undetermined instance $\mathbf{e}$ of dimension $n$ |
| Output: | YES, if there is a determinization $\mathbf{e}'$ of $\mathbf{e}$ such that all completions of $\mathbf{e}'$ are positive, and NO otherwise |

It turns out that DETERMINIZATIONALLPOS is intimately related to PARTIALALLPOS:

**Lemma 1.** *Let $\mathcal{C}$ be any class of models. Then* EVAL(PARTIALALLPOS, $\mathcal{C}$) *can be solved in polynomial time if and only if* DETERMINIZATIONALLPOS($\mathcal{C}$) *can also be solved in polynomial time.*

*Proof.* We prove both directions as separate claims for an arbitrary class of models $\mathcal{C}$.

**Claim 2.** *If* EVAL(PARTIALALLPOS, $\mathcal{C}$) *can be solved in polynomial time then* DETERMINIZATIONALLPOS($\mathcal{C}$) *can also be solved in polynomial time.*

*Proof.* Assume that EVAL(PARTIALALLPOS, $\mathcal{C}$) can be solved in polynomial time for $\mathcal{C}$. Then, consider an instance $(\mathcal{M}, \mathbf{e})$ of DETERMINIZATIONALLPOS, and let $n$ be the dimension of said instance. From $\mathbf{e}$, we build three partial instances $\mathbf{e}_x, \mathbf{e}_y, \mathbf{e}_z$ in the following way:

- $\mathbf{e}_x$ is a determinization of $\mathbf{e}$ such that every undetermined component of $\mathbf{e}$ is replaced by $\bot$ in $\mathbf{e}_x$.

- $\mathbf{e}_y$ is a partial instance that has a 1 in every component where $\mathbf{e}$ has $\bot$, and $\bot$ in every other component.

- $\mathbf{e}_z$ is a partial instance that has a 0 in every component where $\mathbf{e}$ has $\bot$, and $\bot$ in every other component.

We now claim that $\mathcal{M} \models$ PARTIALALLPOS($\mathbf{e}_x, \mathbf{e}_y, \mathbf{e}_z$) if and only if $(\mathcal{M}, \mathbf{e})$ is a positive instance of DETERMINIZATIONALLPOS.

Indeed, assume first that $\mathcal{M} \models$ PARTIALALLPOS($\mathbf{e}_x, \mathbf{e}_y, \mathbf{e}_z$), and let be $\mathbf{e}_u, \mathbf{e}_v, \mathbf{e}_w$ be their witnesses. Trivially, $\mathbf{e}_u$ is a partial instance for which every completion is positive. Note that because $\mathcal{M} \models \mathbf{e}_x \subseteq \mathbf{e}_u$ and the definition of $\mathbf{e}_x$, we have that the defined components of $\mathbf{e}_u$ and $\mathbf{e}$ match. It only remains to see that if $\mathbf{e}[i] = \bot$ for some $1 \leq i \leq n$, then $\mathbf{e}_u[i] = \bot$ as well. Assume to the contrary that for some $i$ it happens that $\mathbf{e}[i] = \bot$ but $\mathbf{e}_u[i] \neq \bot$. If $\mathbf{e}_u[i] = 0$, then $\mathbf{e}_v[i] = 0$, as $\mathcal{M} \models \mathbf{e}_u \subseteq \mathbf{e}_v$. But $\mathbf{e}_v[i] = 0$ contradicts the fact that $\mathbf{e}_y[i] = 1$ (by construction) as $\mathcal{M} \models \mathbf{e}_y \subseteq \mathbf{e}_v$. Similarly, if $\mathbf{e}_u[i] = 1$, then $\mathbf{e}_w[i] = 1$, as $\mathcal{M} \models \mathbf{e}_u \subseteq \mathbf{e}_w$. But $\mathbf{e}_w[i] = 1$ contradicts the fact that $\mathbf{e}_z[i] = 0$ (by construction) as $\mathcal{M} \models \mathbf{e}_z \subseteq \mathbf{e}_w$.

For the other direction, assume $(\mathcal{M}, \mathbf{e})$ is a positive instance of DETERMINIZATIONALLPOS, and let $\mathbf{e}'$ be the determinization of $\mathbf{e}$ that serves as a witness. We claim that $\mathbf{e}_u := \mathbf{e}'$ is a witness for $\mathcal{M} \models$ PARTIALALLPOS($\mathbf{e}_x, \mathbf{e}_y, \mathbf{e}_z$). Indeed, it is trivial that $\mathcal{M} \models \mathbf{e}_x \subseteq \mathbf{e}_u$ as both $\mathbf{e}_x$ and $\mathbf{e}_u$ are determinization of $\mathbf{e}$, but $\mathbf{e}_x$ replaced undetermined components by $\bot$. It is also clear that $\mathcal{M} \models$ ALLPOS($\mathbf{e}_u$), as all completions of $\mathbf{e}'$ are positive by definition. Then, let $\mathbf{e}_v$ be the completion of $\mathbf{e}_u$ that replaces every $\bot$ component of $\mathbf{e}_u$ with 1. Let $\mathbf{e}_w$ be defined analogously but replacing $\bot$ with 0. It is then easy to check that

$$\mathcal{M} \models (\mathbf{e}_y \subseteq \mathbf{e}_v \wedge \mathbf{e}_u \subseteq \mathbf{e}_v) \wedge (\mathbf{e}_z \subseteq \mathbf{e}_w \wedge \mathbf{e}_u \subseteq \mathbf{e}_w)$$

and thus $\mathcal{M} \models$ PARTIALALLPOS($\mathbf{e}_x, \mathbf{e}_y, \mathbf{e}_z$), which is enough to conclude the proof. $\square$

**Claim 3.** *If* DETERMINIZATIONALLPOS($\mathcal{C}$) *can be solved in polynomial time then* EVAL(PARTIALALLPOS, $\mathcal{C}$) *can also be solved in polynomial time.*

*Proof.* Assume that DETERMINIZATIONALLPOS($\mathcal{C}$) can be solved in polynomial time. Then, let $(\mathcal{M}, \mathbf{e}_x, \mathbf{e}_y, \mathbf{e}_z)$ be an input of EVAL(PARTIALALLPOS, $\mathcal{C}$), and let $n = \dim(\mathcal{M})$.

First, we claim that if for some $1 \leq i \leq n$ it happens that $\mathbf{e}_y[i] \neq \bot \neq \mathbf{e}_x[i]$ or $\mathbf{e}_z[i] \neq \bot \neq \mathbf{e}_x[i]$, then we can trivially deduce that $(\mathcal{M}, \mathbf{e}_x, \mathbf{e}_y, \mathbf{e}_z)$ is a negative instance of EVAL(PARTIALALLPOS, $\mathcal{C}$). Indeed, if $\mathbf{e}_y[i] \neq \bot$, and $(\mathcal{M}, \mathbf{e}_x, \mathbf{e}_y, \mathbf{e}_z)$ were to be a positive instance, then there would exists witnesses $\mathbf{e}_u, \mathbf{e}_v, \mathbf{e}_w$, which would hold the following properties:

1. $\mathbf{e}_u[i] = \mathbf{e}_x[i]$, as $\mathcal{M} \models \mathbf{e}_x \subseteq \mathbf{e}_u$ and $\mathbf{e}_x[i]$ is assumed to not be $\bot$.

2. $\mathbf{e}_v[i] = \mathbf{e}_y[i]$, as $\mathcal{M} \models \mathbf{e}_y \subseteq \mathbf{e}_v$ if $\mathbf{e}_y[i] \neq \bot$.

3. $\mathbf{e}_w[i] = \mathbf{e}_z[i]$, as $\mathcal{M} \models \mathbf{e}_z \subseteq \mathbf{e}_w$ if $\mathbf{e}_z[i] \neq \bot$.

4. $\mathbf{e}_v[i] = \mathbf{e}_u[i]$, as $\mathcal{M} \models \mathbf{e}_u \subseteq \mathbf{e}_v$ and $\mathbf{e}_u[i] = \mathbf{e}_x[i]$ is assumed to not be $\bot$.

5. $\mathbf{e}_w[i] = \mathbf{e}_u[i]$, as $\mathcal{M} \models \mathbf{e}_u \subseteq \mathbf{e}_w$ and $\mathbf{e}_u[i] = \mathbf{e}_x[i]$ is assumed to not be $\bot$.

Transitively, it would follow if $\mathbf{e}_y[i] \neq \perp$, then $\mathbf{e}_x[i] = \mathbf{e}_y[i]$, and if $\mathbf{e}_z[i] \neq \perp$, then $\mathbf{e}_x[i] = \mathbf{e}_z[i]$, which contradicts the assumption.

Therefore, we can safely assume from now on that, if $\mathbf{e}_x[i] \neq \perp$, then either $\mathbf{e}_y[i] = \perp$ or $\mathbf{e}_y[i] = \mathbf{e}_x[i]$, and the same holds for $\mathbf{e}_z[i]$. We now define $\mathbf{e}$ as an undetermined instance that is equal to $\mathbf{e}_x$ except that it has $\diamondsuit$ in every component where $\mathbf{e}_x$ has $\perp$. We now claim that $(\mathcal{M}, \mathbf{e}_x, \mathbf{e}_y, \mathbf{e}_z)$ is a positive instance of EVAL(PARTIALALLPOS, $\mathcal{C}$) if and only if $(\mathcal{M}, \mathbf{e})$ is a positive instance of DETERMINIZATIONALLPOS.

Indeed, assume that $(\mathcal{M}, \mathbf{e}_x, \mathbf{e}_y, \mathbf{e}_z)$ is a positive instance of EVAL(PARTIALALLPOS, $\mathcal{C}$). Then, it is trivial that its witness $\mathbf{e}_u$ is a determinization of $\mathbf{e}$ with only positive completions. For the other direction, if $(\mathcal{M}, \mathbf{e})$ is a positive instance of DETERMINIZATIONALLPOS with witness $\mathbf{e}'$, then it is easy to see that taking $\mathbf{e}_u := \mathbf{e}_v := \mathbf{e}_w := \mathbf{e}'$ proves that $(\mathcal{M}, \mathbf{e}_x, \mathbf{e}_y, \mathbf{e}_z)$ is a positive instance of EVAL(PARTIALALLPOS, $\mathcal{C}$), as clearly $\mathcal{M} \models \mathbf{e}_x \subseteq \mathbf{e}_u \wedge \text{ALLPOS}(\mathbf{e}_u)$ and also trivially $\mathcal{M} \models \mathbf{e}_u \subseteq \mathbf{e}_v \wedge \mathbf{e}_u \subseteq \mathbf{e}_w$, thus leaving only $\mathcal{M} \models \mathbf{e}_y \subseteq \mathbf{e}_v \wedge \mathbf{e}_z \subseteq \mathbf{e}_w$ to justify, which we do simply by using the previous fact that if $\mathbf{e}_y[i] \neq \perp$ for some $i$, $\mathbf{e}_x[i] = \mathbf{e}_y[i]$, from which we know that $\mathbf{e}_v[i] = \mathbf{e}_y[i]$,as $\mathcal{M} \models \mathbf{e}_x \subseteq \mathbf{e}_u \subseteq \mathbf{e}_v$. The same reasoning justifies that $\mathcal{M} \models \mathbf{e}_z \subseteq \mathbf{e}_w$

$\square$

The lemma follows directly from the combination of both claims. $\square$

It is easy to see that the same proof applies to PARTIALALLNEG and DETERMINIZATIONALLNEG. We now restate the main theorem of this section and proceed to prove it.

**Theorem 2.** *For every class $\mathcal{C}$ of models, the following conditions are equivalent: (a)* EVAL($\varphi, \mathcal{C}$) *can be solved in polynomial time for each query $\varphi$ in $\exists$FOIL$^+$; (b)* EVAL(PARTIALALLPOS, $\mathcal{C}$) *and* EVAL(PARTIALALLNEG, $\mathcal{C}$) *can be solved in polynomial time.*

*Proof.* The fact that $(a)$ implies $(b)$ is trivial as PARTIALALLPOS and PARTIALALLNEG can be written in $\exists$FOIL$^+$ as shown in the body of the paper. It remains to prove that $(b)$ implies $(a)$. The proof is an extension of the proof of Proposition 1. As it is again constructive and technical, let us first present a sketch. We assume unary predicates EXISTSNEG, EXISTSPOS, that trivially allow for expressing ALLPOS and ALLNEG.

**Sketch of proof** Assume that the input formula is of the form $\varphi = \exists x_1, \cdots, \exists x_k \psi(x_1, \ldots, x_k)$, with $\psi$ quantifier-free, and let $\mathcal{M}$ be the input model with $\dim(\mathcal{M}) = n$. Our algorithm will try to construct a valuation $\mathbf{e}_1, \ldots, \mathbf{e}_k$ of the variables of $\varphi$ such that $\mathcal{M} \models \psi(\mathbf{e}_1, \ldots, \mathbf{e}_k)$, and if this fails, it will be certain that no satisfying valuation exists. In order to do so, the algorithm starts taking $\mathbf{e}_1, \ldots, \mathbf{e}_k$ as undetermined instances, and in particular it starts setting $\mathbf{e}_1 = \mathbf{e}_2 = \cdots = \mathbf{e}_k = \diamondsuit^n$. Then, as $k$ is a fixed constant, the algorithm can afford to guess which unary predicates of $\exists$FOIL$^+$ will be satisfied by each $\mathbf{e}_i$, and also all the containments $\mathbf{e}_i \subseteq \mathbf{e}_j$ that hold. Note that some of such guesses might be inconsistent, as for example, they could fail to respect the transitive property of $\subseteq$, or guess that an $\mathbf{e}_i$ will hold both POS and EXISTSNEG, which is not possible either. Inconsistent guesses are simply discarded. As only constantly many guesses exists, the complicated part of the algorithm is: given a consistent guess, check if it is possible to determinize all instances $\mathbf{e}_1$ through $\mathbf{e}_k$ while respecting the guess. One can show that the complicated cases are captured by the DETERMINIZATIONALLPOS and DETERMINIZATIONALLNEG problems, which because of Lemma 1 are solvable in polynomial time given condition $(b)$.

We assume as well the input formula has no free-variables, as it complicates the exposition without adding combinatorial insight. Let $V = \{\boldsymbol{x}_1, \ldots, \boldsymbol{x}_k\}$ be the variables mentioned in $\varphi$, of which there is only a constant number as $\varphi$ is fixed. Let $E = \{\mathbf{e}_1, \ldots, \mathbf{e}_k\}$ be their corresponding undetermined instances, as the proof sketch suggests. Also, let $\mathcal{M}$ be the input model, and let $n = \dim(\mathcal{M})$. For each element in the domain of $\mathfrak{A}_\mathcal{M}$, that is, for each tuple in $\{0, 1, \perp\}^n$, we define its *type* as the set of unary predicates of $\exists$FOIL$^+$ that it satisfies when interpreted over $\mathfrak{A}_\mathcal{M}$. Note that not all sets of unary predicates are possible types, as for example no tuple can satisfy the set $\{\text{POS}, \text{EXISTSNEG}\}$. Let $\mathcal{T}$ be the set of types that are possible, which is of course a fixed set independent of $\mathcal{M}$. We will guess the type of each instance $\mathbf{e} \in E$. More formally, we can iterate over all type assignments $\tau : E \to \mathcal{T}$ as there is only fixed number of them. Similarly, we can define a *containment assignment* $\gamma$ as an assignment of all the ordered pairs $(\mathbf{e}_i, \mathbf{e}_j)$ to $\{0, 1\}$, with the meaning that $\mathbf{e}_i \subseteq \mathbf{e}_j$ iff $\gamma(\mathbf{e}_i, \mathbf{e}_j) = 1$.

Such an assignment is said to be *possible* only if it holds the properties of a partial order. Given a possible containment assignment $\gamma$, we can interpret it as a pair of sets

$$P = \{(\mathbf{e}_i, \mathbf{e}_j) \mid \gamma(\mathbf{e}_i, \mathbf{e}_j) = 1\} \quad ; \quad N = \{(\mathbf{e}_i, \mathbf{e}_j) \mid \gamma(\mathbf{e}_i, \mathbf{e}_j) = 0\}$$

Note as well that there is a constant number of possibilities for the pair $P, N$. Because the formula $\varphi$ is existential, if there is an determinization of $E$ that models $\varphi$, then there is a pair $(\tau, \gamma = (P, N))$ where $\tau$ is a possible type assignment and $\gamma = (P, N)$ is a possible containment assignment, such that $E$ is *consistent* with both $\tau$ and $\gamma$. More precisely, $E$ is consistent with $\tau$ and $\gamma$ iff:

- For every $\mathbf{e} \in E$ and every unary predicate $\rho$,

$$\rho \in \tau(\mathbf{e}) \iff \mathbf{e} \in \rho^{\mathfrak{A}_{\mathcal{M}}}$$

- For every pair $\mathbf{e}_i, \mathbf{e}_j \in E$,

$$\gamma(\mathbf{e}_i, \mathbf{e}_j) = 1 \iff (\mathbf{e}_i, \mathbf{e}_j) \in \subseteq^{\mathfrak{A}_{\mathcal{M}}}$$

We can afford to iterate over the constantly many pairs $(\tau, \gamma)$, and for each pair $(\tau, \gamma)$ it is trivial to decide whether $\varphi$ gets satisfied under said assignments (simply by replacing every atomic term in $\varphi$ by the value assigned to it by $\tau$ or $\gamma$). Therefore, in order to prove the whole theorem, it is enough to design a polynomial time algorithm that decides whether there determinization of $E$ that is a consistent with a given pair $(\tau, \gamma)$. More precisely, proving the next claim will be enough to conclude our proof.

**Claim 4.** *Given a pair $(\tau, \gamma = (P, N))$, one can check in polynomial time whether there is a determinization of $E$ that is consistent with $(\tau, \gamma)$.*

*Proof.* First, as the desired determinization $E$ must be consistent with $N$, it must hold that for every fact $(\mathbf{e}_i, \mathbf{e}_j) \in N$, there is an index $1 \leq k \leq n$ such that $\mathbf{e}_i[k] \neq \bot$ and $\mathbf{e}_i[k] \neq \mathbf{e}_j[k]$. We can afford to guess, for each of the constantly many facts $(\mathbf{e}_i, \mathbf{e}_j) \in N$, an index $k$ and the values of $\mathbf{e}_i[k], \mathbf{e}_j[k]$, that certify the fact. Also, for every $\mathrm{FULL} \notin \tau(\mathbf{e})$, we can guess a component $\mathbf{e}[k] = \bot$. After said guesses have been made, we can assume a set $F$ of guessed facts of the form $\mathbf{e}[k] = \alpha$, with $\alpha \in \{0, 1, \bot\}$. Then, for every fact in $F$ of the form $\mathbf{e}[k] = \beta$, with $\beta \in \{0, 1\}$, we include in $F$ all facts of the form $\mathbf{e}'[k] = \beta$ for every $\mathbf{e}'$ such that $(\mathbf{e}, \mathbf{e}') \in P$. Also, for every fact in $F$ of the form $\mathbf{e}[k] = \bot$, we include $\mathbf{e}'[k] = \bot$ for every $\mathbf{e}'$ such that $(\mathbf{e}', \mathbf{e}) \in P$. As any determinization of $E$ respecting $F$ will at least be consistent with $N$, it remains only to check whether there is an interpretation $E$ respecting to $F$ that is consistent with $\tau$ and $P$.

If $F$ fully determines some predicates that a certain instance $\mathbf{e} \in E$ must satisfy, for example because $F$ contains facts $\mathbf{e}[k] = \beta \in \{0, 1\}$ for all $1 \leq k \leq n$ and thus we know that $\mathbf{e}$ must be a full instance, we can reject immediately if $\mathrm{FULL} \notin \tau(\mathbf{e})$. Therefore, we can safely assume this is not the case, and that $\tau$ is not directly contradicted by $F$. Let us now interpret $P$ as a directed acyclic graph $G$ obtained in the following way: (i) create a node for every instance $\mathbf{e} \in E$, (ii) create an edge $\mathbf{e} \to \mathbf{e}'$ iff $(\mathbf{e}, \mathbf{e}') \in P$, (iii) collapse strongly connected components to a single node. Note that, as strongly connected components before the last step correspond to instances that must be equal, we can think of them as a single instance, because forcefully $\tau$ must assign the same to each of them. We can now view our problem as that of determinizing every node in a DAG $G$, in such a way that the containment dictated by the graph is satisfied, and so is $\tau$.

If $G$ has multiple connected components (it will only have a constant number of them), it is easy to see that we can simply make the check for each of them separately, and return that the instance is positive if every connected component holds the check. This is because different connected components do not share instances $\mathbf{e}$, and thus a determinization of a connected component is always compatible with the determinization of another connected component. As a consequence, our problem is now even smaller; we need to show that it is possible to determine in polynomial time if the undetermined components in each node of a given connected DAG $G$ can be assigned values that are consistent with given assignments $\tau$ and $P$.

We now show a direct simple algorithm for this problem:

1. Choose an arbitrary topological ordering $\phi$ of $G$.

2. Iterate over the nodes according to $\phi$, and for each node $\mathbf{e}$ do the next step.

3. If FULL $\in \tau(\mathbf{e})$, go to step 4., otherwise go to 5.

4. We either have POS $\in \tau(\mathbf{e})$ or EXISTSNEG $\in \tau(\mathbf{e})$, but not both. On the first case, solve the DETERMINIZATIONALLPOS problem with input $(\mathcal{M}, \mathbf{e})$ and determinize $\mathbf{e}$ accordingly. On the second case, solve the DETERMINIZATIONALLNEG problem. Take $\mathbf{e}$ to be the next node according to $\phi$ and go back to 3. If there is no next node, go to 6.

5. We either have EXISTSPOS $\in \tau(\mathbf{e})$ or EXISTSNEG $\in \tau(\mathbf{e})$, or both. If both, assign every undetermined component of $\mathbf{e}$ to $\bot$, as that only gives more room for completions of $\mathbf{e}$ to be both positive and negative. If only EXISTSPOS $\in \tau(\mathbf{e})$, solve the DETERMINIZATIONALLPOS problem with input $(\mathcal{M}, \mathbf{e})$. If only EXISTSNEG $\in \tau(\mathbf{e})$, solve the DETERMINIZATIONALLNEG problem. Then, propagate each value 0 or 1 that was assigned to an undetermined component of $\mathbf{e}$ to its successors in $G$. Take $\mathbf{e}$ to be the next node according to $\phi$ and go back to 3. If there is no next node, go to 6.

6. Now that nodes have no undetermined components, check that every fact dictated by $\tau$ is true for the values that have been determined. If all the facts are correctly satisfied, return Yes, otherwise return No.

It is clear that, if the preceding algorithm returns Yes, then it is correct, as it has a concrete determinization consistent with $\tau$, and it must be consistent with $P$ as every undetermined component that is assigned 0 or 1 its propagated to the successors in the graph. It only remains to justify that it is correct when it returns No. Assume, looking for a contradiction, that the algorithm returns No but there actually exists a determinization $B$ of $E$ that is consistent with $\tau$ and $P$. Let $A$ be the determinization that the algorithm tested in step 6, and let $i$ be the first node according to $\phi$, the choice of the algorithm in step 1, such that $A(\phi_i) \neq B(\phi_i)$. Such an index must exists because $A$ must differ from $B$. Among all determinizations that are consistent with $\tau$ and $P$, let $B'$ be the one that maximizes the index $i$ of its first difference with $A$. Then, let $\mathbf{e}$ be $i$-th node according to $\phi$, and thus the first node where $A$ and $B'$ differ. If FULL $\in \tau(\mathbf{e})$, then the algorithm determinized $\mathbf{e}$ according to step 4. If POS $\in \tau(\mathbf{e})$, the algorithm determinized $\mathbf{e}$ according to the algorithm for DETERMINIZATIONALLPOS, and thus if $B(\mathbf{e})$ is effectively positive, then $A(\mathbf{e})$ must also be, by Lemma 1 and the theorem hypothesis. Therefore, the inconsistency between $A$ and $\tau$ is not created by $A(\mathbf{e})$. But then, as $\mathbf{e}$ is full, it cannot have any successors in $G$, and thus if we let $B'' := B'$ except for $B''(\mathbf{e}) := A(\mathbf{e})$, then $B''$ must also be consistent with $\tau$, which contradicts the maximality of $i$. The case in which EXISTSNEG $\in \tau(\mathbf{e})$ is analogous.

It remains to see the case where FULL $\notin \tau(\mathbf{e})$. Assume $j$ is the first component for which $A(\mathbf{e})[j] \neq B'(\mathbf{e})[j]$. If $A(\mathbf{e})[j] = \bot$, then it must be the case that both EXISTSPOS $\in \tau(\mathbf{e})$ and EXISTSNEG $\in \tau(\mathbf{e})$. Note that every completion of $B'(\mathbf{e})$ is also a completion of $A(\mathbf{e})$ and thus if $B'(\mathbf{e})$ is consistent with $\tau$, then so is $A(\mathbf{e})$. This implies the inconsistency in $\tau$ must appear later in $\phi$, and thus we can again take $B''$ equal to $B'$ except for $B''(\mathbf{e}) := A(\mathbf{e})$ which will contradict the maximality of $i$.

If $A(\mathbf{e})[j] = 1$ or $A(\mathbf{e})[j] = 0$, it must be the case that EXISTSPOS $\in \tau(\mathbf{e})$ but EXISTSNEG $\notin \tau(\mathbf{e})$, or vice-versa. This means that determinizations of $\mathbf{e}$ must have either all positive completions or all negative completions. Again because of Lemma 1 and the theorem hypothesis, $A(\mathbf{e})$ must hold ALLPOS or ALLNEG if it was possible to determinize $\mathbf{e}$ in that way, which is the case because $B(\mathbf{e})$ does so. Then note that by taking $B''$ which is equal to $B'$ except that it $B''(\mathbf{e}')[j] = A(\mathbf{e})[j]$ for every successor $\mathbf{e}'$ of $\mathbf{e}$ (itself included), we get an assignment that must also be consistent with $\tau$, as EXISTSNEG $\notin \tau(\mathbf{e})$ implies that EXISTSNEG does not hold for any of the successors of $\mathbf{e}$ either. Moreover, no fact of the form FULL can be broken either, as for every non-full variable we already included in $F$ a guess of an undefined component for it. Thus $B''$ is consistent with $\tau$, and it either contradicts the maximality of $i$ or the minimality of $j$. Having explored all possible cases of failure, we can conclude that the algorithm is correct, and as it is clearly polynomial, we finish the proof of this claim.

$\qquad\qquad\qquad\qquad\qquad\qquad\qquad\qquad\qquad\qquad\qquad\qquad\qquad\qquad\qquad\qquad\square$

As the preceding claim has been proved, and there are constantly many pairs $(\tau, \gamma)$ to consider, there is a polynomial time algorithm for the whole problem.

□

In order to finish this section, we restate and prove Proposition 2.

**Proposition 2.** *The problems* EVAL(PARTIALALLPOS, Ptron) *and* EVAL(PARTIALALLNEG, Ptron) *can be solved in polynomial time.*

*Proof.* Based on Lemma 1, it is enough to show that the problems DETERMINIZATIONALLPOS(Ptron) and DETERMINIZATIONALLNEG(Ptron) can be solved in polynomial time. Let us focus on the case of DETERMINIZATIONALLPOS, as the other case is analogous. Thus, an input instance consists of a perceptron $\mathcal{M} = (w, t)$ of dimension $n$, and an undetermined instance $\mathbf{e}$ of dimension $n$. A polynomial time algorithm follows directly from the next claim.

**Claim 5.** $(\mathcal{M}, \mathbf{e})$ *a* YES *instance of* DETERMINIZATIONALLPOS(Ptron) *if and only if the following equation holds*

$$\left( \sum_{i, \mathbf{e}[i] \in \{0,1\}} w_i \mathbf{e}[i] \right) + \left( \sum_{i, \mathbf{e}[i]=\perp} \min(0, w_i) \right) + \left( \sum_{i, \mathbf{e}[i]=\Diamond} \max(0, w_i) \right) \geq t$$

*Proof of Claim 4.* For the forward direction, assume $(\mathcal{M}, \mathbf{e})$ a YES instance, and let $\mathbf{e}'$ the determinization of $\mathbf{e}$ such that all its completions are positive under $\mathcal{M}$. In particular, consider the completion $\mathbf{e}^*$ such that if $\mathbf{e}'[i] = \perp$, then $\mathbf{e}^*[i] = \min(0, w_i)$, and $\mathbf{e}^*[i] = \mathbf{e}'[i]$ otherwise. The fact that this completion is positive means that

$$\left( \sum_{i, \mathbf{e}'[i] \in \{0,1\}} w_i \mathbf{e}'[i] \right) + \left( \sum_{i, \mathbf{e}'[i]=\perp} \min(0, w_i) \right) \geq t$$

Now, the components in $\mathbf{e}'$ can be separated according to whether they were determined or not in $\mathbf{e}$ already:

$$\left( \sum_{i, \mathbf{e}[i] \in \{0,1\}} w_i \mathbf{e}[i] \right) + \left( \sum_{i, \mathbf{e}[i]=\perp} \min(0, w_i) \right) +$$
$$\left( \sum_{i, \mathbf{e}[i]=\Diamond, \mathbf{e}'[i] \in \{0,1\}} w_i \mathbf{e}'[i] \right) + \left( \sum_{i, \mathbf{e}[i]=\Diamond, \mathbf{e}'[i]=\perp} \min(0, w_i) \right) \geq t$$

By noting that $\max(0, w_i) \geq w_i \mathbf{e}'[i]$ and $\max(0, w_i) \geq \min(0, w_i)$ we have that

$$\left( \sum_{i, \mathbf{e}[i]=\Diamond} \max(0, w_i) \right) \geq \left( \sum_{i, \mathbf{e}[i]=\Diamond, \mathbf{e}'[i] \in \{0,1\}} w_i \mathbf{e}'[i] \right) + \left( \sum_{i, \mathbf{e}[i]=\Diamond, \mathbf{e}'[i]=\perp} \min(0, w_i) \right)$$

and thus we conclude simply by combining the three previous equations. For the backward direction, assume the equation holds, and let us define the determinization $\mathbf{e}^\star$ such that

$$\mathbf{e}^\star[i] \begin{cases} \mathbf{e}[i] & \text{if } \mathbf{e}[i] \neq \Diamond \\ 1 & \text{if } \mathbf{e}[i] = \Diamond \text{ and } w_i \geq 0 \\ 0 & \text{otherwise.} \end{cases}$$

Note that this implies that if $e[i] = \Diamond$ then $\mathbf{e}^\star[i] w_i = \max(0, w_i)$. Now let $\mathbf{e}'$ be any completion of $\mathbf{e}^\star$, and we aim to prove that

$$\sum_i w_i \mathbf{e}'[i] \geq t$$

By construction, we have that

$$\sum_i w_i \mathbf{e}'[i] = \left( \sum_{i, \mathbf{e}[i] \in \{0,1\}} w_i \mathbf{e}[i] \right) + \left( \sum_{i, \mathbf{e}[i]=\Diamond} \max(0, w_i) \right) + \left( \sum_{i, \mathbf{e}[i]=\perp} w_i \mathbf{e}'[i] \right)$$

But

$$\left(\sum_{i,\mathbf{e}[i]=\perp} w_i \mathbf{e}'[i]\right) \geq \left(\sum_{i,\mathbf{e}[i]=\perp} \min(0, w_i)\right)$$

and thus

$$\sum_i w_i \mathbf{e}'[i] \geq \left(\sum_{i,\mathbf{e}[i]\in\{0,1\}} w_i \mathbf{e}[i]\right) + \left(\sum_{i,\mathbf{e}[i]=\perp} \min(0, w_i)\right) + \left(\sum_{i,\mathbf{e}[i]=\diamond} \max(0, w_i)\right)$$

which is at least $t$ by hypothesis. Therefore, any completion $e'$ is positive, which concludes the proof. $\qquad\square$

$\square$

In order to make the previous result more meaningful, we show explicitly that the class of perceptrons is not tractable for unrestricted FOIL.

**Proposition 4.** *The problem of deciding whether a model is biased [16], expressible in FOIL through the formula* BIASEDMODEL*, is* NP-*hard for the class of perceptrons.*

*Proof.* In order to show hardness we will reduce from the subset sum problem, which is well known to be NP-hard. Recall that the subset sum problem consists on, given natural numbers $s_1, \ldots, s_n, k \in \mathbb{N}$, to decide whether there is a subset $S \subseteq \{1, \ldots, n\}$ such that $\sum_{i \in S} s_i = k$. Let us proceed with the reduction. Based on a subset sum instance $s_1, \ldots, s_n, k$, we create a perceptron with $n$ unprotected features (that we assume to have indices 1 through $n$) with associated weights $s_1, \ldots, s_n$ and a single protected feature, with index $n+1$ and weight 1. Given the described weights, let $\mathcal{M}$ be the resulting perceptron that has those weights and bias[1] $-k-1$. The following claim is enough to establish the reduction.

**Claim 6.** *The perceptron $\mathcal{M}$ is biased if and only if $s_1, \ldots, s_n, k$ is a positive instance of the subset sum problem.*

For the forward direction, consider $\mathcal{M}$ to be biased. That means there are instances $\mathbf{e}_1$ and $\mathbf{e}_2$ such that $\mathcal{M}(\mathbf{e}_1) \neq \mathcal{M}(\mathbf{e}_2)$, that differ only on the $n+1$-th feature, as it is the only protected one. Assume wlog that $\mathcal{M}(\mathbf{e}_1) = 1$ and $\mathcal{M}(\mathbf{e}_2) = 0$, by swapping the variables if it is not already the case. This implies $\mathbf{w} \cdot \mathbf{e}_1 \geq k+1$ and $\mathbf{w} \cdot \mathbf{e}_2 < k+1$. As $\mathbf{w} \cdot \mathbf{e}_1 \geq \mathbf{w} \cdot \mathbf{e}_2$, and $\mathbf{e}_1$ differs from $\mathbf{e}_2$ only on the $n+1$-th feature, it must hold that $\mathbf{e}_1[n+1] = 1$ and $\mathbf{e}_2[n+1] = 0$, as $w_{n+1} = 1$. Let $P$ be the set of unprotected features of $\mathbf{e}_1$ (and thus $\mathbf{e}_2$) that are set to 1. Then we can write $\mathbf{w} \cdot \mathbf{e}_1 = \left(\sum_{i \in P} s_i\right) + 1$, as each weight $w_i$ was chosen to be equal to $s_i$. Then when considering $\mathbf{e}_1$ we have that $\left(\sum_{i \in P} s_i\right) + 1 \geq k+1$ and, by considering $\mathbf{e}_2$, that $\left(\sum_{i \in P} s_i\right) < k+1$, from which we deduce that $\sum_{i \in P} s_i = k$. We have found a subset of $\{s_1, \ldots, s_n\}$ that adds up to $k$, which is enough to conclude the forward direction of the proof. For the backward direction, consider an arbitrary set $P \subseteq \{1, \ldots, n\}$ such that $\sum_{i \in P} s_i = k$. It is then easy to verify that the instance $\mathbf{e}_1$ that has a 1 in every feature whose index belongs to $P$, a 1 in the $n+1$-th feature, and 0 on the rest, is a positive instance of $\mathcal{M}$. Furthermore, $\mathbf{e}_2$ that differs from $\mathbf{e}_1$ only in the $n+1$-th feature can be checked to be a negative instance. As we have found a pair of instances that differ only on protected features, and yet have opposite classifications, the model $\mathcal{M}$ must be biased. $\qquad\square$

We now restate and prove Proposition 3.

**Proposition 3.** *Let $\mathcal{C}$ be* OBDD *or* DTree. *The problems* EVAL(PARTIALALLPOS, $\mathcal{C}$) *and* EVAL(PARTIALALLNEG, $\mathcal{C}$) *are* NP-*hard.*

*Proof.* . It is enough to prove that hardness holds already for the class of ordered decision trees (i.e., DTree ∩ OBDD). Moreover, based on Lemma 1, it is enough to show that the problems DETERMINIZATIONALLPOS and DETERMINIZATIONALLNEG are NP-hard for ordered decision trees. We focus on the case of DETERMINIZATIONALLPOS, as the other one is analogous.

---

[1] Recall that the bias of a perceptron has nothing to do with the notion of bias that relates to fairness.

We do this by reducing from 3-SAT. Indeed, let $\varphi$ be a formula in 3CNF, with $m$ clauses and $n$ variables. Let us assume that $m = 2^k$ for some integer $k$, as otherwise one can simply add $2^{\lceil \log_2 m \rceil} - m \in O(m)$ clauses consisting of new fresh variables. Then, create an ordered decision tree $\mathcal{T}$ of dimension $k + n$ in the following way:

- The first $k$ features are labeled $a_1, \ldots, a_k$, and then $n$ features corresponding to the variables of $\varphi$ are labeled $x_1, \ldots, x_n$. The features in $\mathcal{T}$ are ordered

$$a_1 \preceq a_2 \preceq \cdots \preceq a_k \preceq x_1 \preceq x_2 \preceq \cdots \preceq x_n.$$

- Start creating $\mathcal{T}$ by building a complete binary ordered tree over the features $a_1, \ldots, a_k$, where the root has label $a_1$, and all nodes at distance $i$ from the root have label $a_{i+1}$. The last layer of said tree consists exactly of $2^{k-1}$ nodes labeled $a_k$.

- For each clause $C_i$ $(1 \le i \le 2^k)$ create an ordered (according to the ordering described above) decision tree $\mathcal{T}_i$ equivalent to said clause. Note that as each clause mentions exactly 3 variables, each of the $\mathcal{T}_i$ can be built in constant time from clause $C_i$.

- Let $L$ be the set of $2^{k-1}$ nodes labeled with $a_k$ in $\mathcal{T}$. Let $\ell_1, \cdots, \ell_{2^{k-1}}$ be any ordering of $L$, and to node $\ell_i$ connect $\mathcal{T}_{2i-1}$ with an edge labeled $0$ and $\mathcal{T}_{2i}$ with an edge labeled $0$.

Note that this construction can trivially be performed in polynomial time. Now build an undetermined instance $\mathbf{e}$ of dimension $k + n$, where $\mathbf{e}[i] = \bot$ for $1 \le i \le k$ and $\mathbf{e}[i] = \Diamond$ otherwise. We claim that there exists a determinization $\mathbf{e}'$ of $\mathbf{e}$ such that all its completions are positive, if and only if, $\varphi$ is satisfiable. Indeed, assume first that such a determinization $\mathbf{e}'$ exists. Then, define $\mathbf{e}''$ in the following way:

$$\mathbf{e}''[i] = \begin{cases} \mathbf{e}[i] & \text{if } \mathbf{e}[i] \neq \Diamond \\ \mathbf{e}'[i] & \text{if } \mathbf{e}[i] = \Diamond \text{ and } \mathbf{e}'[i] \in \{0, 1\} \\ 0 & \text{otherwise.} \end{cases}$$

As by construction $\mathbf{e}' \subseteq \mathbf{e}''$ it must also hold that all completions of $\mathbf{e}''$ are positive. Moreover, note that $\mathbf{e}''[i] = \bot$ for all $1 \le i \le k$ and $\mathbf{e}''[i] \in \{0, 1\}$ for all $k + 1 \le i \le k + n$. We build an $\sigma$ of variables of $\varphi$ based on $\mathbf{e}''$ by setting variable $x_i$ to $\mathbf{e}''[k + i]$, for $1 \le i \le n$. Now, we claim that $\sigma$ is a satisfying assignment. Indeed, to see that $\sigma$ satisfies clause $C_i$, consider the completion $\mathbf{e}^*$ of $\mathbf{e}''$ that sets the features $a_1, \ldots, a_k$ in such a way that the path of $\mathbf{e}^*$ over $\mathcal{T}$ arrives to $\mathcal{T}_i$. As $\mathbf{e}^*$ is a positive instance of $\mathcal{T}$, it must be a positive instance of $\mathcal{T}_i$, and thus by construction $\sigma$ satisfies $C_i$.

For the other direction, let $\sigma$ be a satisfying assignment to $\varphi$, and build the determinization $\mathbf{e}^\star$ such that $\mathbf{e}^\star[k + i] = \sigma(x_i)$ for $1 \le i \le n$. As $\sigma$ satisfies every clause, $\mathbf{e}^\star$ is a positive instance of every $\mathcal{T}_i$, and thus any completion of $\mathbf{e}^\star$ is a positive instance of $\mathcal{T}$. $\qquad\square$

## F   Proof of Theorem 3

In order to make the proof more readable, let us first prove a lemma about simple operations over $k$-COBDDs.

**Lemma 2.** *The following operations can be performed in polynomial time:*

- *(Negation) Given a $k$-COBDD $\mathcal{M}$ of dimension $n$, compute a $k$-COBDD $\neg\mathcal{M}$ over the same ordering of variables, such that $\neg\mathcal{M}(\mathbf{e}) = 1 - \mathcal{M}(\mathbf{e})$ for every instance $\mathbf{e}$ of dimension $n$.*

- *(Disjunction) Given $k$-COBDDs $\mathcal{M}_1$ and $\mathcal{M}_2$, of dimension $n$, with a common linear ordering $<$ on the set $\{1, \ldots, n\}$ compute a COBDD $\mathcal{M} := \mathcal{M}_1 \vee \mathcal{M}_2$ over the same linear ordering $<$, and width at most $2k$, such that $\mathcal{M}(\mathbf{e}) = \max(\mathcal{M}_1(\mathbf{e}), \mathcal{M}_2(\mathbf{e}))$ for every instance $\mathbf{e}$ of dimension $n$.*

- *(Conjunction) Given $k$-COBDDs $\mathcal{M}_1$ and $\mathcal{M}_2$, of dimension $n$, with a common linear ordering $<$ on the set $\{1, \ldots, n\}$ compute a COBDD $\mathcal{M} := \mathcal{M}_1 \wedge \mathcal{M}_2$ over the same linear ordering $<$, and width at most $2k$, such that $\mathcal{M}(\mathbf{e}) = \min(\mathcal{M}_1(\mathbf{e}), \mathcal{M}_2(\mathbf{e}))$ for every instance $\mathbf{e}$ of dimension $n$.*

*Proof.* Negation is trivial, it suffices to interchange the labels **true** and **false** in every leaf of $\mathcal{M}$. Disjunction and Conjunction follow the classical algorithm for OBDDs by Bryant [7], and in what follows we argue that width is no more than doubled. Let us introduce some notation; the inductive structure of OBDDs allows us to say that $\mathcal{M}_1$ has a root node labeled with $r \in \{1, \ldots, n\}$ connected to OBDDs $\mathcal{M}_1^0$ and $\mathcal{M}_1^1$ by edges labeled with 0 and 1 respectively, which we denote as $\mathcal{M}_1 = (r, \mathcal{M}_1^0, \mathcal{M}_2^0)$. Analogously, let $\mathcal{M}_2 = (r, \mathcal{M}_2^0, \mathcal{M}_2^1)$, as $\mathcal{M}_1$ and $\mathcal{M}_2$ share the ordering $<$ and are complete, their root must have the same label. Then, for op $\in \{\text{Conjunction}, \text{Disjunction}\}$, Bryant's algorithm inductively computes operations according to the following equation

$$\text{op}(\mathcal{M}_1, \mathcal{M}_2) = \left(r, \text{op}(\mathcal{M}_1^0, \mathcal{M}_2^0), \text{op}(\mathcal{M}_1^1, \mathcal{M}_2^1)\right). \tag{7}$$

Let us use notation $\mathcal{M} \to j$ for the number of nodes in $\mathcal{M}$ labeled with $j$. We are now ready to prove a stronger claim by induction on $n$, the dimension of the models, from which the lemma trivially follows.

**Claim 7.** *Let $\mathcal{M}_1$ and $\mathcal{M}_2$ be COBDDs of dimension $n$ with a common ordering $<$, and let $j$ be any label in $\{1, \ldots, n\}$. Then Bryant's algorithm guarantees that*

$$\text{op}(\mathcal{M}_1, \mathcal{M}_2) \to j \leq (\mathcal{M}_1 \to j) + (\mathcal{M}_2 \to j).$$

*Proof of Claim 7.* If $n = 1$, the claim is trivial, so we assume $n > 1$. Trivially, $\text{op}(\mathcal{M}_1, \mathcal{M}_2) \to r = 1$, so the claim is also trivial for $j = r$. We now examine the general case of $j \neq r$, for which we will use the inductive hypothesis of $n - 1$. Based on Equation 7, we have that

$$\text{op}(\mathcal{M}_1, \mathcal{M}_2) \to j = (\text{op}(\mathcal{M}_1^0, \mathcal{M}_2^0) \to j) + (\text{op}(\mathcal{M}_1^1, \mathcal{M}_2^1) \to j)$$

Note that $\text{op}(\mathcal{M}_1^0, \mathcal{M}_2^0)$ and $\text{op}(\mathcal{M}_1^1, \mathcal{M}_2^1)$ are COBDDs of dimension $n - 1$, as they do not include label $r$. Then, by inductive hypothesis, we have that

$$(\text{op}(\mathcal{M}_1^0, \mathcal{M}_2^0) \to j) + (\text{op}(\mathcal{M}_1^1, \mathcal{M}_2^1) \to j) \leq (\mathcal{M}_1^0 \to j) + (\mathcal{M}_2^0 \to j) + (\mathcal{M}_1^1 \to j) + (\mathcal{M}_2^1 \to j)$$

But by definition,

$$\mathcal{M}_i \to j = (\mathcal{M}_i^0 \to j) + (\mathcal{M}_i^1 \to j) \qquad (i \in \{1, 2\}).$$

By combining the three previous equations, we get the desired result:

$$\text{op}(\mathcal{M}_1, \mathcal{M}_2) \to j \leq (\mathcal{M}_1 \to j) + (\mathcal{M}_2 \to j).$$

$\square$

We are now ready to finish the proof of the lemma. As for a model $\mathcal{M}$ of dimension $n$ we have that $\text{width}(\mathcal{M}) = \max_{1 \leq j \leq n}(\mathcal{M} \to j)$, it follows from Claim 7 that

$$\text{width}\left(\text{op}(\mathcal{M}_1, \mathcal{M}_2)\right) \leq \max_{1 \leq j \leq n}(\mathcal{M}_1 \to j) + (\mathcal{M}_2 \to j) \leq \text{width}(\mathcal{M}_1) + \text{width}(\mathcal{M}_2)$$

which concludes the proof. $\square$

We now state a lemma of Capelli and Mengel [8, 9] that will be used in our proof, but before let us introduce appropriate notation. Given a COBDD $\mathcal{M}$ of dimension $n$, and a set $S \subseteq \{1, \ldots, n\}$ we define $\exists_S \mathcal{M}$ as a COBDD of dimension $n - |S|$, such that for every instance $\mathbf{e}$ of dimension $n - |S|$, we have that $\exists_S \mathcal{M}(\mathbf{e}) = 1$ iff there is an instance $\mathbf{e}'$ of dimension $n$ that holds both:

- $\mathcal{M}(\mathbf{e}') = 1$
- Let the list $t_1, \ldots, t_{n-|S|}$ correspond to $\{1, \ldots, n\} \setminus S$ in increasing order. Then $e'[t_i] = e[i]$, for every $i \in \{1, \ldots, n - |S|\}$.

For example, if $\mathcal{M}$ is a model of dimension 3 equivalent to $(x_1 \land x_2) \lor (x_2 \land \neg x_3)$, then if we take $S = \{1, 3\}$, $\exists_S \mathcal{M}$ is equivalent to $x_2$, as if $x_2$ is true, then there exists values of $x_1$ and $x_3$ that satisfy $\mathcal{M}$ (namely 1 and 0) respectively, whereas if $x_2$ is false, then no values of $x_1$ and $x_3$ will help satisfy $\mathcal{M}$.

**Lemma 3** (Lemma 1, [8]). *Fix an integer $k > 0$. Given a COBDD $\mathcal{M}$ of dimension $n$ and width $k$, and a set $S \subseteq \{1, \ldots, n\}$, one can compute a COBDD $\exists_S \mathcal{M}$ of width at most $2^k$ in polynomial time.*

We can define $\forall_S \mathcal{M}$ as $\neg\exists_S\neg\mathcal{M}$, and thus the previous lemma applies as well to $\forall_S\mathcal{M}$.

In our case, however, partial instances can have three possible values: $0, 1$, and $\perp$. Therefore, we define *Complete Ordered Ternary Decision Diagrams* (COTDDs) analogously to COBDDs but with nodes having three outgoing edges labeled with $0, 1, \perp$. Note that, given a COBDD $\mathcal{M}$ of dimension $n$, we can build in polynomial time a COTDD $\mathcal{M}^3$ of dimension $n$ such that for every partial instance $\mathbf{e}$ of dimension $n$, $\mathcal{M}^3(\mathbf{e}) = 1$ iff $\mathbf{e}$ is a full instance and $\mathcal{M}(\mathbf{e}) = 1$. This can be done by first creating a path $P$ of nodes according to the underlying order of $\mathcal{M}$, starting from the second label in the ordering, such that each node is connected to the next one by its three outgoing edges, and the last one is connected to a leaf labeled **false**. Then, start by $M^3 := M$, and to each node labeled $u$ in $M^3$, connect it with its outgoing $\perp$ edge to the node labeled with the successor of $u$ in $P$.

We now state the equivalent lemmas for COTDDs.

**Lemma 4.** *The following operations can be performed in polynomial time:*

- *(**Negation**) Given a COTDD $\mathcal{M}$ of dimension $n$ and width $k$, compute a COTDD $\neg\mathcal{M}$ of width $k$ and the same ordering of variables, such that $\neg\mathcal{M}(\mathbf{e}) = 1 - \mathcal{M}(\mathbf{e})$ for every instance $\mathbf{e}$ of dimension $n$.*

- *(**Disjunction**) Given COTDDs $\mathcal{M}_1$ and $\mathcal{M}_2$ of width at most $k$, of dimension $n$, with a common linear ordering $<$ on the set $\{1, \ldots, n\}$ compute a COTDD $\mathcal{M} := \mathcal{M}_1 \vee \mathcal{M}_2$ over the same linear ordering $<$, and width at most $3k$, such that $\mathcal{M}(\mathbf{e}) = \max(\mathcal{M}_1(\mathbf{e}), \mathcal{M}_2(\mathbf{e}))$ for every instance $\mathbf{e}$ of dimension $n$.*

- *(**Conjunction**) Given COTDDs $\mathcal{M}_1$ and $\mathcal{M}_2$ of width at most $k$, of dimension $n$, with a common linear ordering $<$ on the set $\{1, \ldots, n\}$ compute a COTDD $\mathcal{M} := \mathcal{M}_1 \wedge \mathcal{M}_2$ over the same linear ordering $<$, and width at most $3k$, such that $\mathcal{M}(\mathbf{e}) = \min(\mathcal{M}_1(\mathbf{e}), \mathcal{M}_2(\mathbf{e}))$ for every instance $\mathbf{e}$ of dimension $n$.*

*Proof.* The case of negation is exactly as before. For disjunction and conjunction the proof works in the same way but by considering that if $\mathcal{M}_1 = (r, \mathcal{M}_1^0, \mathcal{M}_1^1, \mathcal{M}_1^\perp)$ and $\mathcal{M}_2 = (r, \mathcal{M}_2^0, \mathcal{M}_2^1, \mathcal{M}_2^\perp)$, then the following equations hold:

$$\mathrm{op}(\mathcal{M}_1, \mathcal{M}_2) = \left(r, \mathrm{op}(\mathcal{M}_1^0, \mathcal{M}_2^0), \mathrm{op}(\mathcal{M}_1^1, \mathcal{M}_2^1), \mathrm{op}(\mathcal{M}_1^\perp, \mathcal{M}_2^\perp)\right) \tag{8}$$

$$\mathcal{M}_i \to j = (\mathcal{M}_i^0 \to j) + (\mathcal{M}_i^1 \to j) + (\mathcal{M}_i^\perp \to j) \qquad (i \in \{1,2\}, \forall j \neq r, 1 \leq j \leq \dim(\mathcal{M})) \tag{9}$$
$\square$

**Lemma 5** ( Lemma 1, [8])**.** *Fix an integer $k > 0$. Given a COTDD $\mathcal{M}$ of dimension $n$ and width $k$, and a set $S \subseteq \{1, \ldots, n\}$, one can compute a COBDD $\exists_S\mathcal{M}$ of width at most $2^k$ in polynomial time.*

*Proof.* Direct from the proof in [8], as the same construction can be done in the ternary case. $\square$

We are finally ready to restate our theorem and prove it.

**Theorem 3.** *Let $k \geq 1$ and query $\varphi$ in* FOIL. *Then* EVAL$(\varphi, k\text{-COBDD})$ *can be solved in polynomial time.*

*Proof.* We assume that $\varphi$ alternates quantifiers and starts with an existential one without loss of generality as one can trivially enforce this by adding *dummy* variables. Thus, let $\varphi(x_1, \ldots, x_\ell) = \exists x_{\ell+1} \forall x_{\ell+2} \cdots \forall x_{\ell+m-1} \exists x_{\ell+m} \psi(x_1, \ldots, x_{\ell+m})$, where $\psi$ is quantifier-free, and let $\mathcal{M}, \mathbf{e}_1, \ldots, \mathbf{e}_\ell$ be an input of the EVAL$(\varphi, k\text{-COBDD})$ problem. Let $n = \dim(\mathcal{M})$.

Let us introduce a final piece of notation: from a list of partial instances $\mathbf{e}_1, \ldots, \mathbf{e}_p$ of dimension $n$ each, we can define a unique instance $\mathbf{e}_{[1,p]}$ of dimension $pn$ that is simply the concatenation of instances $\mathbf{e}_1, \ldots, \mathbf{e}_p$. More in general, we will use notation $[i, j]$ for $i \leq j$ to denote the set $\{i, \ldots, j\}$. We use as well $\mathbf{e}[i : j]$ with $i < j$ referring to the partial instance $(\mathbf{e}[i], \ldots, \mathbf{e}[j])$ of dimension $j - i + 1$, where $\mathbf{e}$ is a a partial instance of dimension at least $j$.

Now, the proof consists of two parts. First, we will show that based on $\mathcal{M}$ one can build a COTDD $\mathcal{M}'$ of width at most $f(k)$ for a suitable function $f$, such that

$$\mathcal{M} \models \psi(\mathbf{e}_1, \ldots, \mathbf{e}_{\ell+m}) \iff \mathcal{M}'(\mathbf{e}_{[1,\ell+m]}) = 1$$

We will do this by induction over $|\psi|$ in the next claim, but first let us define a linear ordering $\prec$ of $[1, n(\ell + m)]$ as follows. If $\pi(1), \ldots, \pi(n)$ is the ordering of $\mathcal{M}$, then

$$\pi(1) \prec \pi(1) + n \prec \pi(1) + 2n \prec \ldots \prec \pi(1) + (\ell + m - 1)n \prec$$

continued by

$$\pi(2) \prec \pi(2) + n \prec \pi(2) + 2n \prec \ldots \prec \pi(2) + (\ell + m - 1)n \prec$$

and so on, all the way up to

$$\pi(n) \prec \pi(n) + n \prec \pi(n) + 2n \prec \ldots \prec \pi(n) + (\ell + m - 1)n$$

We now formalize the desired claim.

**Claim 8.** *Let $\phi$ be any formula in* FOIL *mentioning a set of variables $\{x_i, \ldots, x_j\} \subseteq \{x_1, \ldots, x_{\ell+m}\}$, that has at most $c$ logical connectives (i.e., $\wedge, \vee, \neg$). Then, we can build in polynomial time a COTDD $\mathcal{M}_\phi$ over the ordering $\prec$, of dimension $n(\ell + m)$ and width at most $f(c, k)$ for a suitable function $f$, such that for any partial instance $\mathbf{e}$ of dimension $n(\ell + m)$*

$$\mathcal{M}_\phi(\mathbf{e}) = 1 \iff \mathcal{M} \models \phi(\mathbf{e}[i : i + n - 1], \ldots, \mathbf{e}[j : j + n - 1])$$

*Proof.* The proof is by induction on $c$. The base cases ($c = 0$) are constructive and relatively involved, so let us start by the inductive cases, where $c > 0$.

- If $\phi = \phi_1 \vee \phi_2$, simply use the inductive hypothesis to build models $\mathcal{M}_{\phi_1}$ and $\mathcal{M}_{\phi_2}$ and then use Lemma 4 to build $\mathcal{M}_\phi := \mathcal{M}_{\phi_1} \vee \mathcal{M}_{\phi_2}$ of width is at most $3\max(\text{width}(\mathcal{M}_{\phi_1}), \text{width}(\mathcal{M}_{\phi_2}))$. It is not hard to see that the resulting model satisfies the desired conditions.

- If $\phi = \phi_1 \wedge \phi_2$, simply use the inductive hypothesis to build models $\mathcal{M}_{\phi_1}$ and $\mathcal{M}_{\phi_2}$ and then use Lemma 4 to build $\mathcal{M}_\phi := \mathcal{M}_{\phi_1} \wedge \mathcal{M}_{\phi_2}$ of width is at most $3\max(\text{width}(\mathcal{M}_{\phi_1}), \text{width}(\mathcal{M}_{\phi_2}))$. It is not hard to see that the resulting model satisfies the desired conditions.

- If $\phi = \neg\phi_1$, simply use the inductive hypothesis to build model $\mathcal{M}_{\phi_1}$ and then use Lemma 4 to build $\mathcal{M}_\phi := \neg\mathcal{M}_{\phi_1}$ of width is at most $\text{width}(\mathcal{M}_{\phi_1})$. It is not hard to see that the resulting model satisfies the desired conditions.

For the base cases let us introduce some notation to facilitate our construction. We will use $P[u, v]$, for $u \prec v$, to mean a path of nodes labeled from $u$ up to $v$ following the ordering $\prec$ and where each node is connected to the next one by its three outgoing edges. Also, if $a$ and $b$ are nodes, $a \xrightarrow{x} b$ with $x \in \{0, 1, \bot\}$ means that $a$'s outgoing edge labeled with $x$ goes to $b$. If $P$ is a path of nodes, then $P \xrightarrow{x} \alpha$ means that the outgoing edge labeled with $x$ of the last node in $P$ goes to $\alpha$. Similarly, $\alpha \xrightarrow{x} P$ means means the outgoing edge labeled with $x$ of $\alpha$ (which could be the last node of a path if $\alpha$ is one) goes to the first node in $P$. We can now prove that the base cases, when $c = 0$, are also satisfied.

- If $\phi = \text{Pos}(x_i)$ for $i \in [1, \ell + m]$, then we build $\mathcal{M}_\phi$ recursively as follows. Assume $\mathcal{M} = (r, \mathcal{M}^0, \mathcal{M}^1)$, and create a path $P_r = P[r, r + (i - 1)n]$. Then create three identical paths $P_{r \to 0} = P_{r \to 1} = P_{r \to f} := P[r + in, \pi(n) + (\ell + m - 1)n]$. Then, add connections $P_r \xrightarrow{\bot} P_{r \to f} \xrightarrow{0,1,\bot} \textbf{false}$, as positive instances have no occurrences of $\bot$. Connect $P_r \xrightarrow{0} P_{r \to 0}$ and $P_r \xrightarrow{1} P_{r \to 1}$. Finally, if we let $R(\mathcal{M}^0), R(\mathcal{M}^1)$ be the results of the recursive procedure applied to $\mathcal{M}^0$ and $\mathcal{M}^1$, respectively, then we connect $P_{r \to 0} \xrightarrow{0,1,\bot} R(\mathcal{M}^0)$ and $P_{r \to 1} \xrightarrow{0,1,\bot} R(\mathcal{M}^1)$. The recursive procedure applied to a leaf will just keep it as such. It is easy to see that this can be done in polynomial time, and it is not hard to see that

the desired equation for $\mathcal{M}_\phi$ is satisfied. Moreover, note that for each node $r$ in $\mathcal{M}$, we introduce three nodes with the same label (namely, when creating the paths $P_{r\to 0}$, $P_{r\to 1}$ and $P_{r\to f}$, noting that nodes in $P_{r\to f}$ can be shared for every $q$ such that $r \prec q$) which implies the width of $\mathcal{M}_\phi$ is no more than $3k$.

- If $\phi = x_i \subseteq x_j$, we build $\mathcal{M}_\phi$ as follows. We first build a path $P_f = [1, \pi(n) + (\ell + m - 1)n] \to \mathbf{false}$. For each $1 \le t \le n$ we will build a gadget that checks that $\mathbf{e}_i[t] = \bot \lor \mathbf{e}_i[t] = \mathbf{e}_i[t]$. The gadget for $\pi(1)$ will be connected to that of $\pi(2)$ and so on, so the ordering $\prec$ is respected. The exact form of each gadget depends on whether $i < j$ or the opposite, as in the first case $t + (i-1)n \prec t + (j-1)n$, for each $1 \le t \le n$, and vice-versa if $j < i$. Consider first the case where $i < j$. Then, for the gadget for $t \in \{1, \ldots, n\}$ we build a path $P^1 := P[\pi(t), \pi(t) + (i-1)n]$, two identical paths $P^2 = P^3 := P[\pi(t) + in, \pi(t) + (j-1)n]$. and $P^4 := P[\pi(t) + jn, \pi(t) + (\ell + m - 1)n]$. We then connect $P^1 \xrightarrow{0} P^2 \xrightarrow{0} P^4$ and $P^1 \xrightarrow{1} P^3 \xrightarrow{1} P^4$, Also, for a label $u$, let us denote $s(u)$ to its successor according to $\prec$, and let $f(s(u))$ be the node in $P_f$ with label $s(u)$. Next, connect $P^1 \xrightarrow{1} P^3 \xrightarrow{0} f(s(\pi(t) + (j-1)n))$ and $P^1 \xrightarrow{0} P^3 \xrightarrow{1} f(s(\pi(t) + (j-1)n))$. If $t = n$, we connect $P^4 \xrightarrow{0,1,\bot} \mathbf{true}$ and otherwise, if $t < n$ and $G_{t+1}$ is the gadget for $t + 1$, we connect $P^4 \xrightarrow{0,1,\bot} G_{t+1}$. The case when $j < i$ is similar, but every gadget for $1 \le t \le n$ checks first the value of the $t$-th feature in the $j$-th variable, and then checks that the $t$-th variable is either $\bot$ or matches the previously mentioned value. We omit the details as they are a trivial modification of the previous construction.

  It is clear that this procedure takes polynomial time, and it is not hard to see that this procedure satisfies the desired condition. Moreover the resulting COTDD has width 3, as $P^2$ and $P^3$ make for two occurrences of the labels appearing in them, and a final one comes from $P_f$.

As each recursive step reduces $c$ by one, and can at most increase the width by a factor of 3, it follows that the width of the resulting model is at most $k3^c$, and thus $f(c, k) = k3^c$ is a suitable function. This concludes the proof of the claim. $\qquad\square$

Now, for the second part of the proof, assume we have already built $\mathcal{M}'$ by using the previous claim noting that as $\psi$ is a fixed formula, $c$ is a fixed constant which implies that the function $f$ of the previous claim depends thus solely on $k$. Let us now state a simpler claim.

**Claim 9.** *For appropriate sets $S_1, \ldots, S_m$ subsets of $[1, n(\ell + m)]$, that can be determined in polynomial time, the following holds:*

$$\mathcal{M} \models \varphi(\mathbf{e}_1, \ldots, \mathbf{e}_\ell) \iff \exists_{S_1} \forall_{S_2} \cdots \forall_{S_{m-1}} \exists_{S_m} \mathcal{M}'(\mathbf{e}_{[1,\ell]}) = 1$$

*Proof.* Trivial by the definition of $\exists_S$ and $\forall_S$ when defining each $S_i$ as $[n(\ell + i - 1) + 1, n(\ell + i)]$. $\qquad\square$

In order to finish the proof, we use the previous claim and simply compute $\mathcal{M}^\star := \exists_{S_1} \forall_{S_2} \cdots \forall_{S_{m-1}} \exists_{S_m} \mathcal{M}'$ by $m$ repeated applications of Lemma 5, and computing two negations for each $\forall_{S_i}$ according to Lemma 4. This results in $\mathcal{M}^\star$ having width at most $f(k) = 2^{2^{2^{\cdot^{\cdot^{\cdot^{2^{3^{|\varphi|k}}}}}}}}$, where the tower has $m$ times the number 2. Then we build the partial instance $\mathbf{e}_{[1,\ell]}$ and finally evaluate $\mathcal{M}^\star(\mathbf{e}_{[1,\ell]})$, which thanks to the previous claim is enough to solve the whole problem. As every part of the algorithm is proven to be correct, and the running time of each component is polynomial, we conclude the whole proof.

$\qquad\square$

# G   Details of the FOIL implementation and the experimental setting

All our code and instructions to run experiments can be found at the following URL

https://github.com/AngrySeal/FOIL-Prototype

For the implementation of the algorithms we used C++ to assure efficiency. For parsing queries we used the ANTLR (v4.9.1) parser generator. Queries can be specified in a straightforward way in plain text by using tokens `Exists` and `ForAll` for quantification, tokens `~`, `^`, `V`, for logical connectives $\neg$, $\wedge$ and $\vee$. respectively, and tokens of the form `x1, x2, x3, u, v, z`, etc. for variables, plus parentheses for stating precedence. Instances mentioned in the queries are written as `(0,1,?,0)` where `?` represents the $\bot$ value (for defining partial instances). Besides that, we used `P( )` for the unary operator $\textsc{Pos}(\cdot)$ and `<=` for the containment $\subseteq$. The following is an example query.

```
Exists x, Exists y, (P(x) V P(y)) ^ (~( x <= y ) ^ ~(y <= (?,?,?,0,1,?,?)))
```

For debugging purposes we implemented a naive evaluation method (126 lines of code) that considers models as black boxes for evaluating $\textsc{Pos}$, and that evaluates a query by testing all possible combinations for the mentioned variables. For the obvious reasons, this implementation is not practical but it is straightforward to prove its correctness. Thus we use it to check the correctness of the evaluation process of the more efficient algorithms.

We implemented versions for the query evaluator for perceptrons (not described in the paper), decision trees and a modification of FBDDs (see Section J for details). For these last two cases, the implementation had 660 lines of code. Trees and BDDs are passed to the implementation in a straightforward JSON format. We checked the correctness by generating a set of random queries over random models and comparing the output of each algorithm against our naive implementation.

## H  Details of the experimental setting

We used Python for the query and models generation process. We generated random queries with the following recursive process. We initially fix the dimension of the queried model, the number of quantified variables allowed to be used in the complete query, and whether they are going to be universally or existentially quantified. We then construct the quantified-free part as follows. When asking for an expression of size $n$, the query generator method chooses a random size $k$ from 1 to $n-1$, then generates two expressions of size $k$ and $n-k$ and joined them choosing either `^` or `V` randomly. The base case is when asking for an expression of size 1 in which case we choose randomly between $P(x)$, $C$ `<=` $x$, $x$ `<=` $C$ and $x$ `<=` $y$, where $C$ represents a random partial instance constructed according to the dimension of the queried model as a tuple using values `0`, `1` and `?`, and $x$ and $y$ represents randomly chosen variables from all of the variables allowed to be used in the query. Every random choice was done with numpy's `default_rng`. Before returning from every recursive call, the method choose randomly whether a negation (`~`) is added in fron of the expression.

For generating the decision-tree models to be queried, we used the Scikit-learn library. We first select an input dimension, and then we generated a random dataset of that dimension. All input data that we generate are random binary tuples and the target value (classification) is a random bit. Then we select the size $N$ for the tree and trained a decision-tree model with $N$ leaves. Finally, we transformed the obtained decision-tree into a binary one in the JSON format that our implementation can consume.

For the experiments shown in Figure 2a and 2b and using the methods described above, we generated 60 random queries and trained 24 random decision trees of different sizes (see Section 6.1). The performance tests were done in a small personal computer: 64-bits, 2.48GHz, Dual Core Intel Celeron N3060 with 2GB of RAM and Linux Mint 20.1 Ulyssa. Even in this modest machine, the evaluation time for random queries and models was extremely short (see Section 6.1). This gives evidence that our methods can be run even for trees and queries of considerable size in a personal machine without the need of a big computer infrastructure.

## I  A high-level language for FOIL

As we described in the body of the paper, we designed a high-level user-friendly syntax tailored for general models with numerical and categorical features. As FOIL does not allow the use of features beyond binary ones, we need to develop a way for binarizing queries and models. We describe the binarization in the next section, and we only describe here the main features of our user-friendly language.

Figure 3 shows examples of queries written for the Student Performance Data Set [29][2]. In the high-level syntax we use the expressions `exists` and `for every` that represents the logical quantification, and tokens `and`, `or`, `not` and `implies` for typical logical connectives. Variables can be any string that is not a reserved keyword. In the examples in Figure 3 we use `student`, `st1` and `st2` as variables.

Our implementation allows for loading a trained model before evaluating queries (see details on model loading below). Whenever a model is loaded the meta information about features and types as well as the classification is also loaded to be interpreted in the queries. A main difference with basic FOIL is that in our high-level syntax we allow for the use of named features. For example in the first query in Figure 3 we use `student.male` to refer to the binary feature `male` of a student instance. Moreover we can refer to different classes (the output of the model) with names. For our example, we trained a binary classifier in which the positive class name is `goodFinalGrades`. Thus, the expression `goodFinalGrades(student)` is equivalent to the `P( )` expression in the basic FOIL implementation. With all this we can intuitively interpret the first query in Figure 3 as asking if having a male gender is enough for the model to make a decision about the final grades.

Besides naming features, our syntax also allows the comparison of with numerical thresholds. In this case we use the typical `<=` and `>` with their natural meaning. For instance, in our example using the Student Performance Data Set, the feature `alcoholWeek` states the level of alcohol consumption during week days, with value $1$ begin low, and value $5$ being high. Thus, the second query in Figure 3 asks if it is possible that the trained model classify a student with a high alcohol consumption during week days (`student.alcoholWeek > 3`) as having good final grades. The comparison with numerical thresholds departs significantly from the base FOIL formalization, and thus it is not trivial how to compile this type of queries into FOIL. We describe the process in the next section.

Finally, in order to have meaningful answers for existential queries of the form $\exists x(\neg\text{Pos}(x))$, that is, asking for the existence of instances that are classified as negative for the model, we implemented the operator `full( )` that essentially requires an instance to be not partial, that is, not using $\bot$ (the formal definition of this property is in Equation 1). The reason for this is that only total instances can be positive in models and since queries are evaluated over the set of all partial instances, the query $\exists x(\neg\text{Pos}(x))$ is trivially true. Having `full( )` as part of the language allows us to more easily deal with this case. For example, the third query in Figure 3 uses `full( )` to ask if there is a student with low alcohol consumption during the week and that is classified as not getting good grades.

All our queries have as possible answer either `YES` or `NO`. It is not difficult to extend our implementation such that, whenever the answer for an existential query is a `YES` then we can provide an instance as witness for that answer. One can similarly provide a witness when the answer for a universal query is a `NO`. This is part of our ongoing work.

Finally, we handcrafted over 20 queries similar to the ones in Figure 3 and tested them over a decision tree with no more than 400 leaves. You can find the complete set of queries that we tested in our companion code.

## J   Binarization of queries and models

The definition of FOIL considers only binary instances (i.e., tuples in $\{0,1\}^n$ for some $n \geq 1$), and consequently, binary classifiers $\mathcal{M} : \{0,1\}^n \rightarrow \{0,1\}$ for some $n \geq 1$. As many real life classification problems involve a combination of categorical and numerical features, this could present a limitation to our approach. However, in this section we show that it is possible to overcome this apparent drawback by binarizing queries and models. A recent article by Choi et al. [12], studies binary encodings for decision trees as well. Our approach is slightly different, as we are concerned as well with the issue of binarizing queries.

Let us define HL-FOIL as a *high-level* equivalent of FOIL. First, we define a *schema* for HL-FOIL as a mapping from *feature names* to *feature types*, which can be either $\mathbb{R}$ or $\mathbb{B} = \{0,1\}$, intuitively meaning that said feature is numerical or Boolean, respectively. We use notation $S(f)$ to obtain the type of a feature by its name. Moreover, if a schema $S$ defines the type of a feature $f$, we say $f \in S$.

For example, consider the following possible schema for the Student Performance Data Set:

---

```
for every student,
    student.male = true
    implies goodFinalGrade(student)

exists student,
    student.alcoholWeek > 3
    and goodFinalGrade(student)

exists student,
    student.alcoholWeek < 2
    and full(student) and not goodFinalGrade(student)

for every student,
    student.alcoholWeekend > 3 and student.alcoholWeek > 3
    implies not goodFinalGrade(student)

exists student,
    (student.alcoholWeekend > 3 or student.alcoholWeek > 3)
    and student.gradePartial2 <= 6 and student.male = false
    and goodFinalGrade(student)

exists st1, exists st2,
    st1.studyTime > 2 and st2.studyTime <= 3
    and goodFinalGrade(st1) and
    full(st2) and not goodFinalGrade(st2)
```

Figure 3: Example queries in high-level syntax

$$S = \{(\text{age}, \mathbb{R}), (\text{alcoholWeek}, \mathbb{R}), (\text{parentsTogether}, \mathbb{B}), \ldots\}.$$

We say a real-valued decision tree $\mathcal{T}$ is compatible with a schema $S$ if each internal node $u \in \mathcal{T}$ holds one of the following conditions:

- **(Numerical)** Node $u$ has label $(f, \tau)$ for some $f \in S, \tau \in \mathbb{R}$, and $S(f) = \mathbb{R}$.
- **(Boolean)** Node $u$ has label $f$ with $f \in S$, and $S(f) = \mathbb{B}$.

Given a schema $S$, the following are atomic HL-FOIL$_S$ formulas:

- POS$(x)$, where $x$ is a variable.
- FULL$(x)$, where $x$ is a variable.
- $(\leq, x, f, \tau)$, where $x$ is a variable, $S(f) = \mathbb{R}$, and $\tau \in \mathbb{R}$.
- $(=, x, f, b)$, where $x$ is a variable, $S(f) = \mathbb{B}$, and $b \in \mathbb{B}$.

Naturally, the domain of HL-FOIL$_S$ consists of functions from feature names $f \in S$ to values in $S(f) \cup \{\bot\}$, which we call instances of HL-FOIL$_S$. Continuing with our running example, the function $\mathbf{e}$ such that $\mathbf{e}(\text{age}) = 19.4, \mathbf{e}(\text{parentsTogether}) = 0, \ldots$ is an instance of HL-FOIL$_S$. The semantics for the atomic formulas FULL$(x)$, $(\leq, x, f, \tau)$, and $(=, x, f, b)$ is naturally defined as one would expect, by checking whether $\mathbf{e}_x(f) \leq \tau$, and $\mathbf{e}_x(f) = b$, respectively. In order to clarify the semantics of POS, we detail how instances of HL-FOIL$_S$ are evaluated by a decision tree.

For a decision tree $\mathcal{T}$ compatible with $S$ and an instance $\mathbf{e}$ of HL-FOIL$_S$, we define $\mathcal{T}(\mathbf{e})$ inductively:

- If $\mathcal{T}$ is a leaf labeled with **true**, then $\mathcal{T}(\mathbf{e}) = 1$, and $\mathcal{T}(\mathbf{e}) = 0$ if the label is **false**.
- If $\mathcal{T}$ has a root labeled with $(f, \tau)$, left sub-tree $\mathcal{T}_0$, and right sub-tree $\mathcal{T}_1$, then $\mathcal{T}(\mathbf{e})$ is defined as follows. If $\mathbf{e}(f) \leq \tau$, then $\mathcal{T}(\mathbf{e}) = \mathcal{T}_1(\mathbf{e})$, and otherwise $\mathcal{T}(\mathbf{e}) = \mathcal{T}_0(\mathbf{e})$.

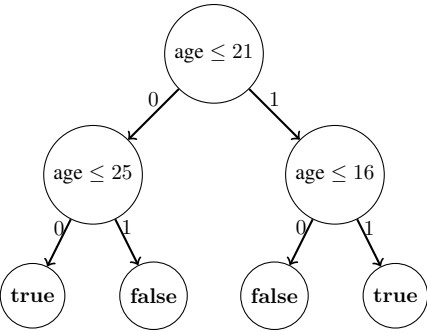

Figure 4: Example of real-valued decision tree. For the sake of clarity, labels are written as $f \leq \tau$ instead of $(f, \tau)$.

- If $\mathcal{T}$ has a root labeled with $f$, left sub-tree $\mathcal{T}_0$, and right sub-tree $\mathcal{T}_1$, then $\mathcal{T}(\mathbf{e})$ is defined as $\mathcal{T}_{f(\mathbf{e})}(\mathbf{e})$.

It is important to stress that Boolean decision trees, in order to prevent inconsistencies, cannot repeat node labels in any path from the root a leaf. For real-valued decision trees we need a stronger requirement to avoid inconsistencies: if $\mathcal{T}$ has a root labeled with $(f, \tau)$, left sub-tree $\mathcal{T}_0$, and right sub-tree $\mathcal{T}_1$, then all nodes labeled with $(f, \tau_0)$ in $\mathcal{T}_0$ must hold $\tau_0 > \tau$, and similarly all nodes labeled with $(f, \tau_1)$ in $\mathcal{T}_1$ must hold $\tau_1 < \tau$.

We are now ready to define a binarization procedure.

**Definition 1.** *A binarization procedure $\mathcal{B}$ is an algorithm that takes: (i) a schema $S$; (ii) an existential formula $\varphi$ in HL-FOIL$_S$ ; (iii) a decision tree $\mathcal{T}$ compatible with $S$, and returns a formula $\psi \in (\exists \text{FOIL} \cup \text{FULL})$ together with a binary model $\mathcal{M}$, such that*

$$\mathcal{T} \models \varphi \iff \mathcal{M} \models \psi.$$

The rest of this section is dedicated to show an efficient binarization procedure $\mathcal{B}$. First, let us show the intuition behind the procedure with a simple example. Figure 4 depicts a real-valued decision tree $\mathcal{T}$ over the schema $S$ of our running example. Note that, although an instance $\mathbf{e}$ can have any real value as $\mathbf{e}(\text{age})$, $\mathcal{T}$ only distinguishes 4 intervals:

$$(-\infty, 16], \ (16, 21], \ (21, 25], \ (25, \infty)$$

Now consider the HL-FOIL$_S$ query:

$$\varphi = \exists x \, \text{Pos}(x) \wedge (\leq, x, \text{age}, 27)$$

Consider now instances $\mathbf{e}_1, \mathbf{e}_2$ such that $\mathbf{e}_1(\text{age}) = 26$ and $\mathbf{e}_2(\text{age}) = 28$. While $\mathcal{T}$ accepts both $\mathbf{e}_1$ and $\mathbf{e}_2$ by traversing the same path, only $\mathbf{e}_1$ is a witness for $\varphi$. This implies that we require a finer partition of the real line into intervals. Namely,

$$\mathcal{I}_{\text{age}} = (-\infty, 16], \ (16, 21], \ (21, 25], \ (25, 27], \ (27, \infty)$$

is a *correct partition* for the tuple $(\mathcal{T}, \varphi, \text{age})$. Based on this, as $|\mathcal{I}_{\text{age}}| = 5$, we will use $5 - 1 = 4$ binary features to encode the age of instances. In particular, the leftmost 1 among those 4 binary features will indicate the interval to which the age of an instance belongs, interpreting that if there is no 1 among the 4 binary features, it belongs to the last interval. This will then allow to do the following compilation from HL-FOIL$_S$ to FOIL:

$$(\leq, x, \text{age}, 27) \rightsquigarrow \neg (x \subseteq (0, 0, 0, 0, \bot, \bot, \ldots))$$

As a FOIL instance $\mathbf{e}$ not having 0 in the first four Boolean features that encode age must have at least a 1 in one of those Boolean features, and thus, it corresponds to a HL-FOIL$_S$ instance whose age lies in one of the first four intervals of $\mathcal{I}_{\text{age}}$, and therefore have age $\leq 27$.

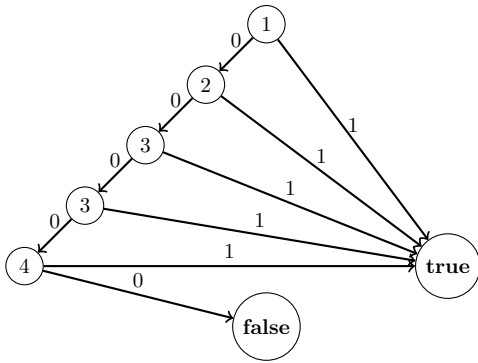

Figure 5: Illustration of $\mathcal{B}(\mathcal{T}, \varphi)$, assuming age is the only feature, and considering a query $\varphi = \exists x \text{Pos}(x)$ that does not mention any threshold $\tau$.

More formally, assume a real-valued decision tree $\mathcal{T}$ and a HL-FOIL$_S$ query $\varphi$. Then, for every feature name $f \in S$ such that $S(f) = \mathbb{R}$, we define its *partition set* as follows:

$$P_f = \{\tau \mid (f, \tau) \text{ labels a node in } \mathcal{T}, \text{ or } (\leq, x, f, \tau) \text{ appears in } \varphi \text{ for some variable } x\}$$

Feature $f$ will be encoded using $|P_f|$ Boolean features. The resulting dimension of instances of HL-FOIL$_S$ compiled into FOIL instances will therefore be $n = \sum_{f \in S} |P_f|$, taking the convention that if $S(f) = \mathbb{B}$ then $|P_f| = 1$.

As $S$ is unordered, but instances of FOIL have an ordering of their features, we choose an arbitrary ordering of features names $f_1, \ldots, f_k$, and we associate to them ranges of Boolean features as follows. To $f_1$ we associate the components in the range $[1, |P_{f_1}|]$, and then for $f_i, i > 1$, we associate $[\ell, \ell + |P_{f_i}| - 1]$, where $\ell$ is end of the range associated to $f_{i-1}$ plus 1.

Therefore comparison of the form $(\leq, x, f, \tau)$, for some variable $x$, will thus be compiled as:

$$(\leq, x, f, \tau) \rightsquigarrow \neg (x \subseteq \mathbf{e}_\tau)$$

where, if $\tau$ is the $i$-th smallest element in $P_f$, then $\mathbf{e}_\tau$ is an instance having 0 in the first $i$ Boolean features associated to $f$, and $\bot$ in the rest.

We now need to binarize the decision tree $\mathcal{T}$ accordingly. We do so by transforming $\mathcal{T}$ into a BDD $\mathcal{B}(\mathcal{T}, \varphi)$ (not necessarily free) that is *almost*-free, in a precise sense that we will detail. If $\mathcal{T}$ was using a single node $u$ to test whether the age of an instance was at most 27, we now require several nodes to test for the different Boolean features encoding the age of said instance. In particular, if age is encoded with Boolean features of indices $[i, \ldots, j]$, and 27 is the $k$-th smallest value in $P_{\text{age}}$, then $\mathcal{B}(\mathcal{T}, \varphi)$ needs to test that there is a 1 in among the features of indices $[i, \ldots, i + k - 1]$. Thus, we create in $\mathcal{B}(\mathcal{T})$ a *gadget* for node $u$ of $\mathcal{T}$ that tests whether

$$\mathbf{e}[i] \vee \mathbf{e}[i + 1] \vee \cdots \vee \mathbf{e}[i + k - 1].$$

Correctness is clear from the construction of the binarization procedure. Figure 5 illustrates $\mathcal{B}(\mathcal{T}, \varphi)$ continuing with the previous example.

While $\mathcal{B}(\mathcal{T}, \varphi)$ is not necessarily an FBDD, we claim that it is close enough to one in what concerns the evaluation of $\exists$FOIL formulas. As the proof of Proposition 1 implies, the only characteristic of models we require for efficient evaluation in the existential fragment is that one can find in polynomial time a determinization of an undetermined instance that is positive for said models. This is clearly not possible for general BDDs, as even checking if there is one positive instance for a BDD is NP-hard [38]. Surprisingly, the same algorithm for FBDDs presented in the proof of Proposition 1 turns out to work for $\mathcal{B}(\mathcal{T}, \varphi)$.

In order to see illustrate why this is true, we consider a more sophisticated example of a real-valued decision tree $\mathcal{T}$, presented in Figure 6. Note that both the gadgets for nodes labeled $(\text{age}, 21)$ and $(\text{age}, 25)$ will use the Boolean features associated to age, and thus the *freeness* property will be broken. The reason the presented algorithm does not work in general BDDs is that a positive leaf could only

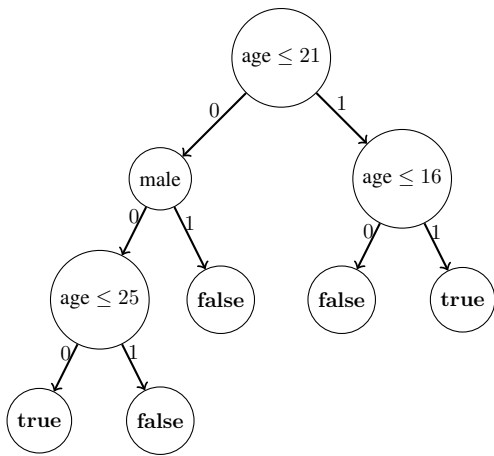

Figure 6: Example of real-valued decision tree. For the sake of clarity, labels are written as $f \leq \tau$ instead of $(f, \tau)$.

be reachable through inconsistent paths, i.e., paths that contain both edges representing that a feature has value 0 and 1. In the case of $\mathcal{B}(\mathcal{T}, \varphi)$, features may appear multiple times in a path from root to leaf, but always as part of different gadgets associated to different nodes in $\mathcal{T}$. This implies that, even if an inconsistent choice is made for a particular Boolean feature of $\mathcal{B}(\mathcal{T}, \varphi)$, an inconsistent path in $\mathcal{B}(\mathcal{T}, \varphi)$ still corresponds as a consistent path in $\mathcal{T}$, because paths in $\mathcal{B}(\mathcal{T}, \varphi)$ translate back to paths in $\mathcal{T}$ by considering if gadgets where exited by failing or succeeding the disjunction they represent.