# OpenReview forum: "Foundations of Symbolic Languages for Model Interpretability"
_NeurIPS.cc/2021/Conference — NeurIPS 2021 Spotlight_

### Official Review · Reviewer_deaJ · 2021-07-08

**Rating:** 7
**Confidence:** 4

**Summary:**

The authors present a logical language, called FOIL, in which
many simple yet relevant interpretability queries can be expressed.
They go on to study the computational complexity
of FOIL queries over two classes of ML models often
deemed to be easily interpretable: decision trees and OBDDs.

The authors show that despite claims of the good level of interpretability
for models such as DTs and OBDDs, there is a simple query in FOIL that is
intractable over them. The authors go on to to define two tractable
fragments of FOIL.

**Limitations And Societal Impact:**

Yes.

**Main Review:**

This is a very well-written and detailed paper. The appendix is
long and contains interesting material. The proof of Theorem 1 is particularly clever.

This field of research appears novel: defining a new language for
interpretability queries, analysis of the complexity of queries when the models are restricted.

I am not very convinced that this is a definitive language which will be
used in the field, since it is basically a low-level language. For example
the definition of the notion MATCH (line 210) is a roundabout way of
saying something quite simple, i.e. that x and y agree on features in S.
However, the authors have implemented a translator from a higher-level
language to their low-level language. So the interest of FOIL is its
simplicity since this helps for proving (in)tractability results.

The theoretical study is restricted to DTs and OBDDs, which leaves open
similar questions for more complex models (such as random forests or d-DNNFs).

line 165: specify whether instances can be partial or not. If they are
necessarily total instances, then the claim that EVAL(phi,CNF) is NP-complete
doesn't make sense.

Minimal Sufficient Reason: perhaps note that this is known by other
authors as PI-explanations or abductive explanations (see, for example,
Shih et al. A symbolic approach to explaining bayesian network classifiers,
IJCAI 18 and Marques-Silva et al. Explaining Naive Bayes and Other Linear
Classifiers with Polynomial Time and Delay, NeurIPS 20).

Line 367: 'a particular subclass of BDDs': specify (in the body of the paper)
exactly which subclass.

Line 371: 'medium size decision trees': what size are we talking about here?

some problems with English:
- legend fig.1: does the requester has a stable job
- line 41: we explain along the paper
- line 196: The following query express this
- line 319: The whole package needs of
- line 330: controlling by three aspects


**Time Spent Reviewing:**

5

---

> ### Author Response · Authors · 2021-08-11
> **Response to Reviewer deaJ**
>
> [Comment] “I am not very convinced that this is a definitive language which will be used in the field, since it is basically a low-level language. For example the definition of the notion MATCH (line 210) is a roundabout way of saying something quite simple, i.e. that x and y agree on features in S. However, the authors have implemented a translator from a higher-level language to their low-level language. So the interest of FOIL is its simplicity since this helps for proving (in)tractability results.”
>
> [Response] We agree with this sentiment; it occurs indeed that in order to have a very minimalistic logic, certain queries or predicates that are simple and natural require sophisticated or non-intuitive queries in the logic. However, as the theoretical analysis shows they are within the expressive power of the language, it is possible to think that an even higher level implementation would provide the user with different forms of syntactic sugar that would then compile to these roundabout ways of saying simple things.
>
> [Comment] “The theoretical study is restricted to DTs and OBDDs, which leaves open similar questions for more complex models (such as random forests or d-DNNFs).”
>
> [Response] Indeed. Many subclasses of decision diagrams, or ensemble models, could be interpretable with respect to FOIL. Studying structural conditions that a class of models needs to respect in order to have efficient evaluation properties is part of our future work. Note that Theorem 2 already provides an interesting characterization of classes of models for which tractability could be extended.
>
> [Comment] “line 165: specify whether instances can be partial or not. If they are necessarily total instances, then the claim that EVAL(phi,CNF) is NP-complete doesn't make sense.”
>
> [Response] Thanks for pointing this out. Indeed line 165 should have said “(partial)”. Interestingly, even if the input is restricted to full instances, one can check that the problem is still NP-hard. This is because the proof of Theorem 1 only requires a partial instance $e$ that has 1s in a subset of the coordinates, and $\bot$ on the rest, and this can be simulated in the restricted problem by giving as input two full instances $e_1$, where $e_1$ has 1s in every feature, and $e_2$ has 1s in the features where $e$ had 1s, and 0s where $e$ had $\bot$. Thus, one can notice that $e$ is the partial instance that has both $e_1$ and $e_2$ as completions  which is largest according to the order induced by $\subseteq$. This  allows to write
> $$
> Q(e_1, e_2) = \exists e, [e \subseteq e_1 \land e \subseteq e_2 \land \forall e' ((e' \subseteq e_1 \land e' \subseteq e_2) \rightarrow e' \subseteq e)] \land P(e)
> $$
>  where $P$ is the query used in the proof of Theorem 1.
>
> [Comment] “Minimal Sufficient Reason: perhaps note that this is known by other authors as PI-explanations or abductive explanations”
>
> [Response] Thanks for the suggestion. In a camera ready version we would add pointers to such papers, as indeed these queries have been studied in the past under different names. It is not our intention to contribute to the multiplicity of names, and we took the Darwiche & Hirth name as we believed it was the most self-explanatory.
>
> We thank you for the detailed suggestions and also for pointing out some typos and some sentences that could improve their clarity.

---

### Official Review · Reviewer_tR2Z · 2021-07-14

**Rating:** 3
**Confidence:** 4

**Summary:**

This paper proposed a query language that can be used to ask yes/no questions of a model. The language was investigated for OBDDs and decision trees.

**Ethical Concerns:**

No issues I saw.

**Limitations And Societal Impact:**

Yes

**Main Review:**

The main problem with this paper is that it claims to be about interpretability, but I can't see how the proposed query language is more interpretable than say the original decision tree or OBDD.  Why is asking Boolean questions helpful for interpretability? It might be obvious to you, but isn't to me. Please give some concrete use cases. (As an example, one use case might be to able to explain why some examples in a test set are given the classification thay were given.) For any complete case, it gives the same answer as the model.

Originality. The paper is moderately original; there is not a lot of novel content. I was surprised that you did not have two sorts of queries: "could" an instance of this query be positive and "must" an instance of this partial query be positive. (I realize the logic can hadle this, but you then don't go and propose what would be a usable language for someone else to use to query a model).

Quality: The quality is reasonable; it just doesn't do vey much.

Clarity: That paper is reasonably clear.

Significance:  This is very low. I can't imagine anyone using this for anything useful.

Given theorem 1, I would think that you would redesign FOIL.  Give a formalism such as a decision tree that is already very interpretable (OBDDs no so much), it seems strange to want to ask complicated queries over them and then claim it is about interpretability.

For a top conference like NeurIPS, there needs to be a concrete result. Average time for simple queries as a function of input dimensions is not a very interesting result. (Why is it so slow?)  If the claims are about interpretability, then the results should be about interpretability. Have other users tried to interpret the results?

I don't think NeurIPS is the right venue for this paper. The style of theorems would be more suited to a conference like KR.

Overall, I am sceptical that  giving yes/no answers is useful for interpretability. You assume that it is true, but never justify it.


**Time Spent Reviewing:**

2

---

> ### Author Response · Authors · 2021-08-11
> **Response to Reviewer tR2Z**
>
> [Comment] “The main problem with this paper is that it claims to be about interpretability, but I can't see how the proposed query language is more interpretable than say the original decision tree or OBDD. Why is asking Boolean questions helpful for interpretability? It might be obvious to you, but isn't to me. Please give some concrete use cases. (As an example, one use case might be to able to explain why some examples in a test set are given the classification thay were given.) For any complete case, it gives the same answer as the model.”
>
> [Response] While we understand that Decision Trees or OBDD are usually deemed as interpretable, several authors have pointed out already that this claim needs to be taken with caution (cf. The Mythos of Model Interpretability). Indeed, while a Decision Tree with 20 leaves can be looked at directly, this is no longer the case for one with thousands of nodes, where simply traversing paths in the tree could be impossible for humans. It is in this context that it becomes helpful to be able to query the model in a declarative manner; asking whether a certain subset of the features fully define the classification of a given instance is something that can be done in FOIL (as the Minimal Sufficient Reason example shows in the paper), and that would hardly be possible for a human to decide by looking at the model directly. It is indeed true that Boolean questions are restrictive, and that a more practical version of a declarative language for interpretability would require the possibility of yielding outputs of different forms. The decision of considering a Boolean output is simply because it allows for a cleaner theoretical analysis, but it is not hard to see that for tractable models (e.g., Decision Trees in the existential fragment) in a query of the form Exists x, Phi(x), one could obtain a satisfying x besides the “Yes” answer with the same algorithm that is used in the appendix to prove tractability.
>
> We acknowledge that the paper would benefit by being much more explicit in this respect.
>
> [Comment] “Given theorem 1, I would think that you would redesign FOIL. Give a formalism such as a decision tree that is already very interpretable (OBDDs no so much), it seems strange to want to ask complicated queries over them and then claim it is about interpretability.”
>
> [Response] We understand the concern, but our opinion is that, while Decision Trees are indeed considered very interpretable, this claim needs to be understood with certain limitations. Indeed, wide and unordered decision trees do not allow to answer certain natural interpretability questions that users could pose. Theorem 1 provides a concrete example; it is computationally hard to tell whether a given partial instance allows for a stable (i.e., robust under small perturbations) completion over a Decision Tree. It has also been shown in the literature [https://proceedings.neurips.cc/paper/2020/hash/b1adda14824f50ef24ff1c05bb66faf3-Abstract.html] that finding minimum size sufficient reasons is NP-hard for Decision Trees. Therefore, we think our paper helps understanding the limits of interpretability for Decision Trees by presenting a counterexample to their general explainability (Theorem 1), while also providing a positive result with respect to a fragment of the logic (Corollary 1), which shows that there is a certain degree of interpretability one can formally prove for Decision Trees by considering our logical yardstick. Note as well that this is not trivial, as for example it is clearly not shared by multilayer perceptrons. We would like to add a paragraph with part of this analysis in the body of the paper for the camera-ready version.
>
> [Comment] “​​For a top conference like NeurIPS, there needs to be a concrete result. Average time for simple queries as a function of input dimensions is not a very interesting result. (Why is it so slow?) If the claims are about interpretability, then the results should be about interpretability. Have other users tried to interpret the results?”
>
> [Response] We respectfully disagree with part of this comment. We believe that our theoretical analysis takes an initial step towards a practical implementation by providing a high-level syntax, an implementation of the algorithm, and testing that it works in a reasonable time on a very low-end personal computer. Note that most queries for Decision Trees with up to 1000 nodes take less than a second to complete, and thus it is reasonable for a user to wait for them on an interactive session of querying the model. We agree however that our proposal would be strengthened by a user study, and this is part of our current work. Note that our paper does not attempt to claim that the proposed language, even in its high level syntax, is ready for practitioners. We acknowledge that several steps are needed in order for it to have a practical impact with ML practitioners. However, we believe our paper sets a firm foundation for declarative languages aiming to help interpretability, and also sets a boundary for them with its intractability results.
>
> While challenging, we appreciate the review and suggestions as we believe it has given us good leads on how to improve the clarity of our paper, and pointed out several things that should be indeed more explicit.

---

### Official Review · Reviewer_oTbW · 2021-07-17

**Rating:** 8
**Confidence:** 3

**Summary:**

This paper introduces a simple logic for examining classifiers. It shows
1) The logic allows defining, in particular, a predicate capturing (minimal)
sufficient reason explanations (i.e., a set of feature values that ensure the
classifier gives a certain classification) and a predicate indicating whether
the classifier makes a decision using a protected set of features.
2) The most natural version of the logic allows defining a predicate that is
NP-hard to evaluate for both decision tree and OBDD classifiers
3) A simple fragment of the logic is adequate to capture the formulas indicated
in (1), and there are two formulas such that if these can be evaluated in
polynomial time for a model class, then the entire fragment is polynomial time
4) The fragment in (3) is polynomial-time decidable for decision trees and OBDDs
(on account of the indicated criterion)
5) A restricted kind of OBDD allows the more general logic to be decided in
polynomial time.

**Limitations And Societal Impact:**

One class of properties that are missing from the logic that is orthogonal to the kind of counting queries supporting SHAP-like attribution that are mentioned in the future work are queries that concern the performance in practice -- e.g., what kind of errors is the model likely to make? These probably would require some kind of probabilistic extension, where queries are evaluated wrt the actual data distribution. This is relatively minor, in that the paper at least indicates that there are *some* natural queries that are not captured.

**Main Review:**

There has been quite a lot of work on methods for interpreting/auditing/etc.
machine learning models. As the paper observes, indeed, it seems inevitable that
there will be a plethora of distinct tasks for different purposes. Actually,
the "no silver bullet" discussion in the opening paragraph really should cite
the influential report by Doshi-Velez and Kim

Doshi-Velez and Kim, Towards A Rigorous Science of Interpretable Machine Learning, arXiv:1702.08608, 2017.

I recognize it is an arXiv report, but it is very well known at this point and
lends credibility to these claims.

Generally speaking, what is needed are tools that permit a user (an auditor,
customer, engineer, etc.) to ensure that the models are meeting the needs of an
application. To my knowledge, most work in this area has focused on evaluating
one particular property, or surveying the kinds of properties these users want
evaluated (mostly work in CHI in that case). As far as I am aware, the tack
taken in this work is original, developing a framework and tools to capture the
whole family of such tasks. I believe this paper could be influential. At a
technical level, the logic and fragments, and the criterion for polynomial-time
evaluation of the fragment, are all interesting contributions in my view.

One may argue that it is a limitation of the framework that it isn't operating
on neural networks. I'm not so worried about this -- it's worthwhile to
capture what can be checked against models that are "known to be interpretable",
before worrying about what can be extended to messy models like neural nets.
It is still helpful to present tools to a user that allow them to explore such
models and validate them against various requirements.

**Time Spent Reviewing:**

4

---

> ### Author Response · Authors · 2021-08-11
> **Response to Reviewer oTbW**
>
> [Comment] ​​”Actually, the "no silver bullet" discussion in the opening paragraph really should cite the influential report by Doshi-Velez and Kim.”
>
> [Response] We agree, this is a good suggestion that we didn’t consider while writing the paper. We were however conscious of such a report, and it is indeed appropriate to cite it.
>
> [Comment] “One class of properties that are missing from the logic that is orthogonal to the kind of counting queries supporting SHAP-like attribution that are mentioned in the future work are queries that concern the performance in practice -- e.g., what kind of errors is the model likely to make? These probably would require some kind of probabilistic extension, where queries are evaluated wrt the actual data distribution. This is relatively minor, in that the paper at least indicates that there are some natural queries that are not captured.”
>
> [Response] We agree. Probabilistic reasoning for explainability is indeed very important, and a logic that can handle such kinds of queries is thus an interesting line of research. Many queries we deal with have a probabilistic counterpart that is not explored in the paper. For example, “minimum/minimal sufficient reasons'' are a deterministic version of queries like min-\delta-relevant sets [https://arxiv.org/abs/2106.00546], which ask for sets of features that, while not fully sufficient for a verdict, make it very likely. Moreover, many formal characterizations of bias are probabilistic (e.g., the probability of acceptance is similar for individuals of different classes with respect to a protected feature). As proposed by the reviewer, querying about the kind of errors a model is more likely to make is also an interesting direction of study. We thank you for the idea.

---

### Official Review · Reviewer_WuQH · 2021-07-21

**Rating:** 7
**Confidence:** 2

**Summary:**

The authors propose a declarative, logic-based language to query machine learning models. The language is tailored to a syntax and semantics that supports queries that explore the behaviour of the machine learning model. For instance, using Pos(e) as an atom that expresses that input e (which might contain undefined arguments) is a positive example, that is, an example classified as positive by the ML model, and using predicates for individual features such as “married” and “has kids”, one could query whether the model under consideration would give a loan to a person who is married and does not have kids.

Given such a logical language, it is natural to ask what the complexity of answering queries in the languages are and to restrict the language in ways that make queries tractable. This is what the authors set out to do in the submitted paper.

**Limitations And Societal Impact:**

Yes

**Main Review:**

The motivation for the problem under consideration and the general writing style is fantastic. Despite its somewhat “exotic” topic for a machine learning conference, the paper is accessible and I enjoyed reading it a lot.

Generally, this is a really nice paper that approaches XAI from the angle of declarative languages.

The major shortcoming (in my opinion) of the paper is the somewhat hidden link between the logical semantics and the operation one has to run using the machine learning model. It would be good to explain early in the paper how a statement like “Pos(e)” would be evaluated given a (real-world) machine learning model. Is it that one would query the ML models with “e” and, based on the classification result, return true/false? Given a query such as “exists x P(x)”, is the idea to query the ML model (in the worst case) an exponential number of times? Is the assumption, in other words, that an ML model can be seen as a function which we can query in constant time? Is it true that the idea is limited to binary covariates (features)? (I assume the answer to the above questions is “yes” but I am not 100% sure). Making these types of connections more explicit, early on in the paper, would build a better bridge between the typical ML readership and the proposed logical formalism.

The authors specify two classes of “models” for which they analyse the query language: decision trees and OBDDs. It would be good to motivate OBDDs a bit more as I have not seen an OBDD used as a machine learning model. I understand that it is a nice formalism that renders certain queries tractable. But what is the relevance to ML here?

I was intrigued by the bias example starting with line 201. This is one definition of bias (a very strong form as far as I can tell) but does not fully cover all situations one might want to call biased. In the end, what one wants is that the protected covariate is independent of the response variable (the model output). Is that satisfied by the definition of “Bias” here? Again, as you can see, I find the connection between the logical approach of the authors and the typical concepts in machine learning sometimes difficult to establish.
Generally, I think this is a strong paper. My worry is that most ML researchers would skip the paper because the link between their “reality”, that is, concepts and terminology, and what is used in the paper is missing or not made clear enough.


**Time Spent Reviewing:**

2 hours

---

> ### Author Response · Authors · 2021-08-11
> **Response to Reviewer WuQH**
>
> [Comment] “The major shortcoming (in my opinion) of the paper is the somewhat hidden link between the logical semantics and the operation one has to run using the machine learning model. It would be good to explain early in the paper how a statement like “Pos(e)” would be evaluated given a (real-world) machine learning model. Is it that one would query the ML models with “e” and, based on the classification result, return true/false?
>
> [Response] This is a good suggestion regarding the presentation of our paper; we indeed agree with making such connections more explicit and earlier on in the paper. As an implementation of FOIL has access to the ML model that the user is trying to interpret, the evaluation of Pos(e) would simply consist of running e through the model. For example, in Figure 1b, the first line of code grants access to the internals of the ML model (e.g., the weights of an MLP) by receiving a file with the model serialized in it. So your understanding in the last sentence is indeed correct.
> [Comment] “Given a query such as “exists x P(x)”, is the idea to query the ML model (in the worst case) an exponential number of times? Is the assumption, in other words, that an ML model can be seen as a function which we can query in constant time?”
>
> [Response] When a query such as “exists x, P(x)” is evaluated, indeed the naive approach would be to try all the exponentially many partial instances and check if at least one of them is accepted; however, this would be prohibitively expensive even for small feature spaces (say, n = 30). While, in general, answering this query over a model might be a computationally intractable task, we know that for several models of interest, such as the ones studied in the paper, it is not.  The algorithms required for polynomial time evaluation of this, and even further classes of queries, over such models work in a completely different way to the naive approach explained above (cf. Appendix): they are based on reasoning over the exponentially-sized set of partial instances in an *implicit* way that avoids any exponential blow up.
>
> Regarding the assumption that “an ML model can be seen as a function which we can query in constant time?”, our paper does not make such an assumption. However, FOIL formulas can only be evaluated in polynomial time over ML models that can be evaluated in polynomial time over any instance (i.e., Given model M and instance e, computing M(e) takes polynomial time). Note that this is not very restrictive, as even more sophisticated models such as multilayer perceptrons can be evaluated in polynomial time for any instance.
>
> [Comment] “Is it true that the idea is limited to binary covariates (features)? (I assume the answer to the above questions is “yes” but I am not 100% sure). Making these types of connections more explicit, early on in the paper, would build a better bridge between the typical ML readership and the proposed logical formalism.”
>
> [Response] The limitation to binary features was a choice in order to simplify the theoretical analysis. One can see, although this is not explicit in the paper, that the same kind of analysis can be performed for features with k possible values for any integer k. Dealing with continuous features is much more complicated, and a general take on it remains as future work. However, over certain kinds of models, such as Decision Trees, it is possible to binarize a continuous model, trained with numerical data, and answer numerical questions about it. Further details on this contribution can be found in Section I of the appendix. We indeed agree that the paper would benefit from stating these ideas even more explicitly and as early in the body as possible.
>
> [Comment] “The authors specify two classes of “models” for which they analyse the query language: decision trees and OBDDs. It would be good to motivate OBDDs a bit more as I have not seen an OBDD used as a machine learning model. I understand that it is a nice formalism that renders certain queries tractable. But what is the relevance to ML here?”
>
> [Response] This is a great question. Indeed OBDDs are not (to the best of our knowledge) commonly used by ML practitioners. Originally we thought it would be easy to evaluate any query over Decision Trees, but we then realized that certain natural queries are hard to evaluate (Theorem 1), which reaffirms the idea that one cannot simply say that Decision Trees are always easily interpretable (cf. The Mythos of Model Interpretability). By looking at which restrictions of Decision Trees would give better properties for evaluating queries over them, we found out that both width and a consistent ordering of features were important, which led us to OBDDs.
>
> [Comment] “This is one definition of bias (a very strong form as far as I can tell) but does not fully cover all situations one might want to call biased.”
>
> [Response] You are right; the definition studied in the paper is arguably one of the most simplistic ones, and we should have made this much more explicit. Indeed many forms of bias are not captured by this definition, and many of those seem to require forms of probabilistic reasoning that are not captured by FOIL, being thus part of our future work.

---

### Author Response · Authors · 2021-08-11
**General comments**

We thank all reviewers for their comments and suggestions. We will first make some general observations here and then address each reviewer directly.

We are excited to see that reviewers share our enthusiasm for this project introducing declarative languages for XAI, as we strongly believe that a solid theoretical foundation is key for achieving a rigorous understanding of the possibilities and limits of explainability.  In fact, three of the reviewers agree that our theoretical contributions are sound, interesting, and novel. We also thank the reviewers for their appreciation of the writing and the general presentation of the paper.

There is, however, a general question that three reviewers share, which is about the gap between our theoretical proposal and the practice of ML. First, we acknowledge that this gap exists, and furthermore, we have taken certain steps in the paper to bridge the said gap: as the Boolean domain in which the logic takes place is indeed restrictive, we extended its use to arbitrary numerical features through a non-trivial process of binarization which is detailed in the appendix. Moreover, several examples of queries over a realistic dataset are presented. Nonetheless, as reviewers correctly pointed out, there are still several steps to take in order to make a declarative language that is usable in practice. We want to stress here that our work aspires to be a solid foundation for said languages, and thus we considered the simplest setting that allows for a meaningful theoretical study. It is this view that has motivated our choice of considering only Boolean features and answers, or defining a logic as minimalistic as possible so more sophisticated queries and tools could be built on top of it. We would like to point out that this practice is a standard in the theoretical foundations of logical languages of different kinds, in areas such as databases, model checking, knowledge representation, concurrency, and many others. We plan to add further detail in a camera-ready version about additions that would be required for the language to be more practical.

We provide detailed responses to each one of the individual reviews below.

---

### Decision · Program_Chairs · 2021-09-27

**Decision:**

Accept (Spotlight)

**Comment:**

The paper attracted significant discussion among reviewers. The reviewers identified the potential of a declarative unifying framework and rigorous theoretical analysis.  The reviewers also highlighted the novelty as a core strength. On the other hand, one reviewer did point out the lack of convincing empirical study.

The consensus among reviewers was that the fact paper drew diverse and strong opinions makes it a great candidate for acceptance since it has the potential to spur such discussions in the community. Given that all the reviewers came to a consensus that the paper be accepted (even though not every reviewer agreed with the objectives of the paper), the job of AC was an easy one.

We hope that the authors will take the comments of all the reviewers very seriously and incorporate them into the final version.